# Inferring mitochondrial and cytosolic metabolism by coupling isotope tracing and deconvolution

Alon Stern[1], Mariam Fokra[2], Boris Sarvin [2], Ahmad Abed Alrahem[2], Won Dong Lee [2], Elina Aizenshtein[2], Nikita Sarvin[2] & Tomer Shlomi [1,2,3] ✉

The inability to inspect metabolic activities within distinct subcellular compartments has been a major barrier to our understanding of eukaryotic cell metabolism. Previous work addressed this challenge by analyzing metabolism in isolated organelles, which grossly bias metabolic activity. Here, we describe a method for inferring physiological metabolic fluxes and metabolite concentrations in mitochondria and cytosol based on isotope tracing experiments performed with intact cells. This is made possible by computational deconvolution of metabolite isotopic labeling patterns and concentrations into cytosolic and mitochondrial counterparts, coupled with metabolic and thermodynamic modelling. Our approach lowers the uncertainty regarding compartmentalized fluxes and concentrations by one and three orders of magnitude compared to existing modelling approaches, respectively. We derive a quantitative view of mitochondrial and cytosolic metabolic activities in central carbon metabolism across cultured cell lines without performing cell fractionation, finding major variability in compartmentalized malate-aspartate shuttle fluxes. We expect our approach for inferring metabolism at a subcellular resolution to be instrumental for a variety of studies of metabolic dysfunction in human disease and for bioengineering.

Metabolic activities are localized in distinct subcellular compartments in Eukaryotic cells. Mitochondrial and cytosolic metabolism play a central role in energy production, biosynthesis, and redox balance. Numerous alterations in these metabolic activities have been associated with human disease. Each subcellular compartment has distinct metabolite pools, pH, energy and redox state, and significantly differ in metabolic flux. Mitochondrial and cytosolic metabolism is tightly interlinked through the transport of metabolic intermediates as well as redox and energy co-factors, which complicates their studying once isolated. Inferring mitochondrial and cytosolic metabolic activities under physiological cellular conditions remains a major open challenge[1].

A variety of methods were proposed for inferring the concentration of metabolites as well as energy and redox cofactor in mitochondria and cytosol[2–9]. The most straightforward approach is performing measurements in isolated mitochondria[4,5], though mitochondrial extraction typically perturbs its metabolism. Recent studies suggested methods for rapid cell fractionation and quenching of metabolism to enable measurements of physiological concentrations; e.g., via immunocapture of epitope-tagged organelles[5,10], nonaqueous fractionation (NAF)[8], or through digitonin-based selective permeabilization of plasma membrane[6]. However, these approaches are technically challenging and may alter the concentration of metabolites having rapid turnover rates. Attempts to measure physiological redox and energy cofactor

[1]Department of Computer Science, Technion—Israel Institute of Technology, 32000 Haifa, Israel. [2]Department of Biology, Technion—Israel Institute of Technology, 32000 Haifa, Israel. [3]Lokey Center for Life Science and Engineering, Technion—Israel Institute of Technology, 32000 Haifa, Israel. ✉e-mail: tomersh@cs.technion.ac.il

ratios were traditionally performed by measuring the cellular concentration of reactants of mitochondrial or cytosolic redox reactions assumed to be in chemical equilibrium[11–16]. More recently, fluorescence protein reporters have been developed to measure NAD+/NADH ratio in the cytosol[17–21] and mitochondria[19,21,22].

The inference of mitochondrial and cytosolic metabolic flux is even more challenging. A most common approach for inferring fluxes is feeding cells with $^{13}$C tracers, measuring metabolite isotopic labeling patterns, and fitting the measurements with model simulations via metabolic flux analysis (MFA)[23–26]. However, the measured isotopic labeling of a metabolite in eukaryotic cell extracts represents a mixture of potentially distinct labeling forms across subcellular compartments. The latter has not been previously accounted for in MFA-based studies of eukaryotic cell metabolism (implicitly assuming similar isotopic labeling forms of metabolites in mitochondria and cytosol), which could markedly bias flux interpretation. Attempts to infer compartmentalized flux via isotope tracing techniques applied to isolated mitochondria may suffer from nonphysiological conditions[4,10,27]. Recently, inference of central carbon metabolic fluxes in cultured cells was demonstrated by performing isotope tracing followed by rapid cell fractionation to infer compartment-specific isotopic labeling of metabolites[6]. The inference of specific compartment-specific fluxes was also demonstrated by utilizing particular isotopic tracers that are metabolized differently in mitochondria and cytosol[28,29], based on the isotopic labeling of secreted metabolites[30], and based on flux balance analysis (FBA) approach[31].

Here, we show how mitochondrial and cytosolic metabolic activities can be inferred based on metabolic measurements performed on intact cells under physiological conditions by combining metabolic modeling and computational deconvolution. We start by showing how compartment-specific redox cofactor ratios can be inferred based on deconvolution of metabolic concentration measurement and thermodynamics. Then, we provide a generic modeling approach for jointly inferring compartmentalized fluxes and concentrations based on deconvolution of isotope tracing measurements coupled with metabolic modeling. We demonstrate the applicability of this approach, deriving a first quantitative view of mitochondrial and cytosolic central energy fluxes and concentrations across human cancer cell lines without cell fractionation that potentially perturbs the cells.

## Results

### Inferring mitochondrial and cytosolic redox cofactor ratios analytically by deconvolution of metabolite concentration measurements

We describe a simple approach for directly inferring redox cofactor concentrations and ratios in mitochondria and cytosol based on whole-cell measurements:

Cytosolic NADP + /NADPH ratio can be estimated based on the thermodynamics of 6-Phosphogluconate dehydrogenase (6PGD) in the cytosol, oxidizing 6-phosphogluconate (6PG) to produce ribulose-5-phosphate (R5P) by reducing NADP+ to NADPH (Fig. 1a). We infer the ratio of forward−backward flux through 6PGD (denoted by $J^+$ and $J^-$, respectively) by feeding cells with [U-$^{13}$C]-glucose and measuring the mass-isotopomer distribution of 6-phosphogluconate (the fraction of the metabolite pool having zero, one, two, etc. isotopic carbons; Fig. 1b). M + 6 6PG (i.e., having 6 isotopic carbons) is produced from fed isotopic glucose through the oxidative pentose-phosphate pathway, while M + 5 6PG is synthesized through reverse 6PGD flux from R5P (incorporating $^{12}$CO$_2$). Hence, under isotopic steady state, mass balance entails that the forward−backward ratio is equal to $\frac{R5P_{m+5}}{6PG_{m+5}}$ ("Methods").

According to the flux-force relationship[32], the ratio of forward-to-backward flux through 6PGD is proportional to the Gibbs free energy of this reaction:

$$\Delta G = - RT \ln(J^+/J^-) \tag{1}$$

Hence, measuring the total cellular concentration of R5P and 6PG (and assuming that they are strictly cytosolic), enables to infer the cytosolic NADP+/NADPH ratio:

$$\Delta G = \Delta G_0 + RT^* \ln \frac{R5P^*CO_2^*NADPH_{CY}}{6PG^*NADP_{CY}} \tag{2}$$

$\Delta G_0$ denotes the standard Gibbs free energy of 6PGD and estimated here based on the group contribution method[33–35]. R and T represent the gas constant and temperature, respectively.

Hence, we get:

$$\frac{NADP+_{CY}}{NADPH_{CY}} = \frac{R5P^*CO_2}{6PG} * \frac{J^+}{J^-} e^{\frac{\Delta G_0}{RT}} \tag{3}$$

The measured total cellular concentration of NADP+ and NADPH represents a mixture of their concentration in mitochondria and cytosol and can be deconvoluted to the compartment-specific concentrations[36]:

$$(1 - \alpha)NADP_{MT} + (\alpha)NADP_{CY} = NADP_{WC} \tag{4}$$

$$(1 - \alpha)NADPH_{MT} + (\alpha)NADPH_{CY} = NADPH_{WC} \tag{5}$$

where $\alpha$ denotes the relative volume of the cytosol out of total cellular volume (Methods). Combining these deconvolution equations with an estimate of cytosolic NADP+/NADPH ratio enables to infer bounds on the mitochondrial and cytosolic concentration of these co-factors (as well as on the mitochondrial cofactor ratio; "Methods").

Applied to cultured HeLa cells, we find that while the whole-cell NADP+/NADPH ratio is 0.1–0.3, the cytosolic ratio is at least sixfold higher (-2–150), while the mitochondrial ratio is not higher than -0.3 (Fig. 1c; showing 95% confidence interval computed based on experimental noise in isotope tracing and concentration measurements; "Methods"). The lower NADP+/NADPH ratio in mitochondria is associated with -2 orders of magnitude higher concentration of NADPH in mitochondria (-1 mM) versus in cytosol (<0.02 mM); while the whole-cell NADPH level is -0.1 mM (Fig. 1d). The estimated cytosolic NADP +/NADPH ratio is consistent with previous measurements performed in iBMK cells using similar thermodynamic considerations[37]. Notably, our estimated NADP+/NADPH ratio is 1–2 order of magnitude higher than previous estimates that were based on simplifying assumptions that the NADP+/NADPH-dependent malic enzyme 1 (ME1)[13,14,16] and isocitrate dehydrogenase 1 (IDH1)[16] are at chemical equilibrium and that the total cellular concentration of reactants in these enzymes match the cytosolic concentrations; which do not hold here (see "Discussion").

A similar approach can be used to estimate compartmentalized NAD+ and NADH levels. We infer the cytosolic NAD+/NADH ratio by analyzing the forward−backward flux of cytosolic lactate dehydrogenase (LDH), feeding cells with [U-$^{13}$C]-lactate, and measuring the total cellular concentration of pyruvate and lactate (Fig. 1e–h). Notably, while pyruvate also exists in mitochondria, the total cellular concentration of pyruvate provides a good estimate of the cytosolic concentration; assuming net flux of cytosolic pyruvate into mitochondria and considering the thermodynamics of the mitochondrial pyruvate carrier (MPC; "Methods").

We find the cytosolic NAD+/NADH ratio >1000 (Fig. 1g), which is at the upper bound of previous estimates for this ratio[13];

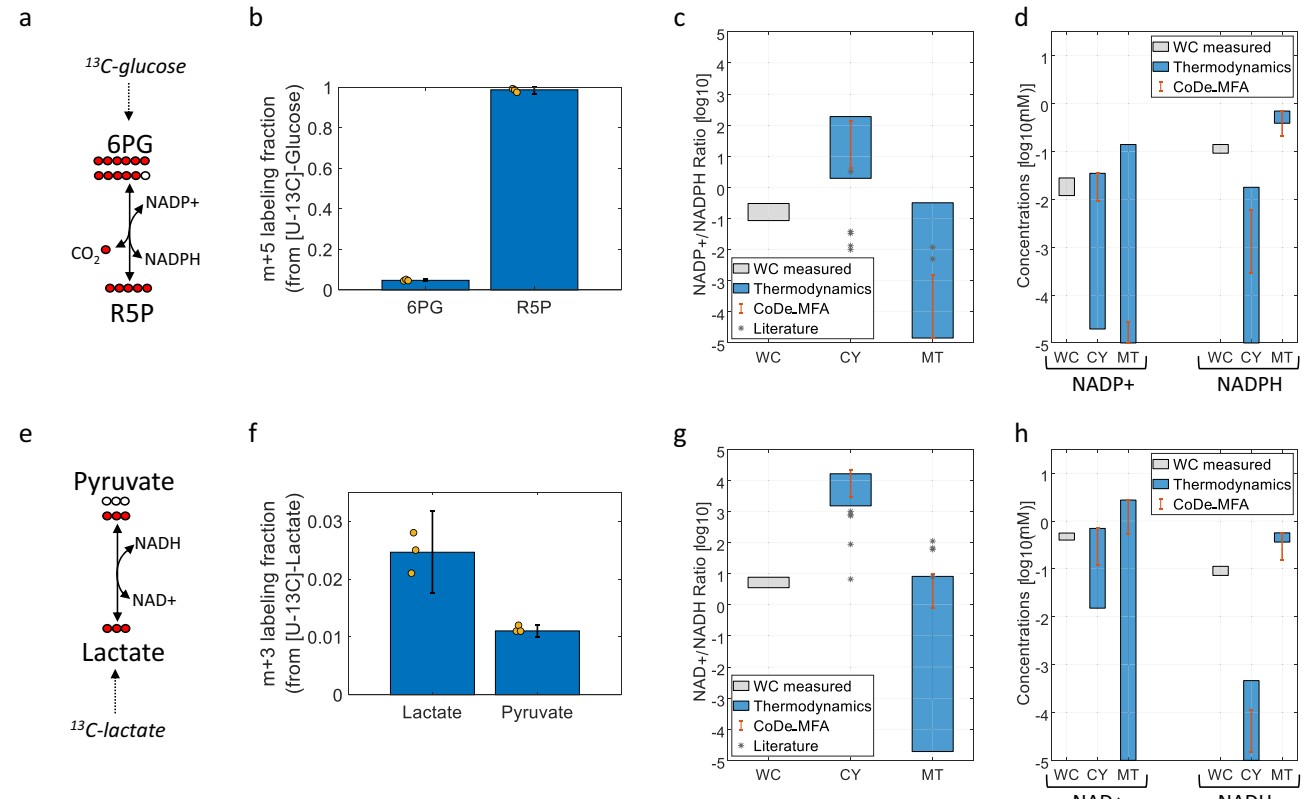

**Fig. 1 | Inferring mitochondrial and cytosolic cofactor concentrations and ratios in HeLa cells. a** A scheme of 6-Phosphogluconate dehydrogenase (6PGD) and reactant isotopic labeling forms when feeding [U-¹³C]-glucose. **b** The steady-state fractional M + 5 labeling of 6-phosphogluconate (6PG) and ribulose-5-phosphate (R5P) when feeding [U-¹³C]-glucose (data are presented as mean values ± SD, n = 3 independent biological replicates). **c**, **d** The measured NADP+/NADPH ratio and NADP+ and NADPH concentrations (gray) and inferred compartmentalized ratio and concentrations based on simple thermodynamics and

deconvolution analysis (blue), and via CODE-MFA (red). Asterisks represent previously published ratios (Supplementary Table S12). **e** A scheme of lactate dehydrogenase (LDH) and reactant isotopic labeling forms when feeding [U-¹³C]-lactate. **f** The fractional M + 3 labeling of lactate and pyruvate when feeding [U-¹³C]-lactate (data are presented as mean values ± SD, n = 3 independent biological replicates). **g**, **h** The measured NAD+/NADH ratio and NAD+ and NADH concentrations (gray) and inferred compartmentalized values based on simple thermodynamics and deconvolution analysis (blue), and via CODE-MFA (red).

obtained based on thermodynamics of LDH, though without accounting for the displacement from chemical equilibrium. Deconvolution of whole-cell NAD+ and NADH measurements suggests that the mitochondrial NAD+/NADH ratio is lower than 10. This is one order of magnitude lower than previous estimates made based on the thermodynamics of beta-hydroxybutyric dehydrogenase[11,14,16], glutamate dehydrogenase[11,14], and malate dehydrogenase[14] (considering similar simplifying assumptions regarding chemical equilibrium of reactions and compartment-specific reactant concentrations as described above). The markedly higher NAD+/NADH ratio in the cytosol than in mitochondria is due to a ~3-order of magnitude lower concentration of NADH in the cytosol than in mitochondria (Fig. 1h). This is consistent with previous measurements performed with a genetically encoded fluorescent sensor for intracellular NADH detection, showing ~300-fold higher concentration of NADH in mitochondria than in cytosol[38].

**A generic approach for inferring mitochondrial and cytosolic metabolite levels and fluxes by computational deconvolution of total cellular concentration and isotope tracing measurements**
Measured metabolite concentrations and isotopic labeling in eukaryotic cell extracts represents a mixture of metabolite pools from different subcellular compartments, which is typically not accounted for in MFA studies. Here, we developed a method, COmpartment-DEconvoluted Metabolic Flux Analysis (CODE-MFA), for inferring

mitochondrial and cytosolic metabolite concentrations and fluxes, explicitly considering that measured metabolite concentrations and isotopic labeling patterns represent average quantities across compartments; and also considering cellular uptake and secretion rates, and biomass growth requirements. The method utilizes optimization to search for the most likely fluxes and metabolite concentrations within a compartmentalized metabolic network model that are optimally consistent with the experimental measurements (Fig. 2a; "Methods"): Simulated metabolite concentrations in mitochondria and cytosol are constrained based on thermodynamic principles. Specifically, the second law of thermodynamics, associating reaction reactant concentrations with the direction of net flux (Fig. 2a; Eq. (I–II)); and the flux-force relationship, associating reactant concentrations with the forward–backward flux ratio (Fig. 2a; Eq. (III)). Convolution of simulated metabolite concentrations in mitochondria and cytosol is matched with the measured concentrations, considering the relative volume of each subcellular compartment (Eq. (2a), Eq. (IV)). Simulated mitochondrial and cytosolic steady-state fluxes (Fig. 2a; Eq. (VI–VII)) were utilized to uniquely determine compartment-specific metabolite isotopic labeling patterns (utilizing Elementary Metabolite Unit[39]). A convolution of the simulated isotopic labeling pattern (i.e., mass-isotopomer distribution) of a metabolite in the two compartments is matched with the measured isotopic labeling pattern (Eq. (2a), Eq. (V); Fig. 2b). Notably, the convolution of the simulated isotopic labeling patterns of a metabolite in mitochondria and cytosol relies on the simulated concentration of the metabolite in these compartments; i.e.,

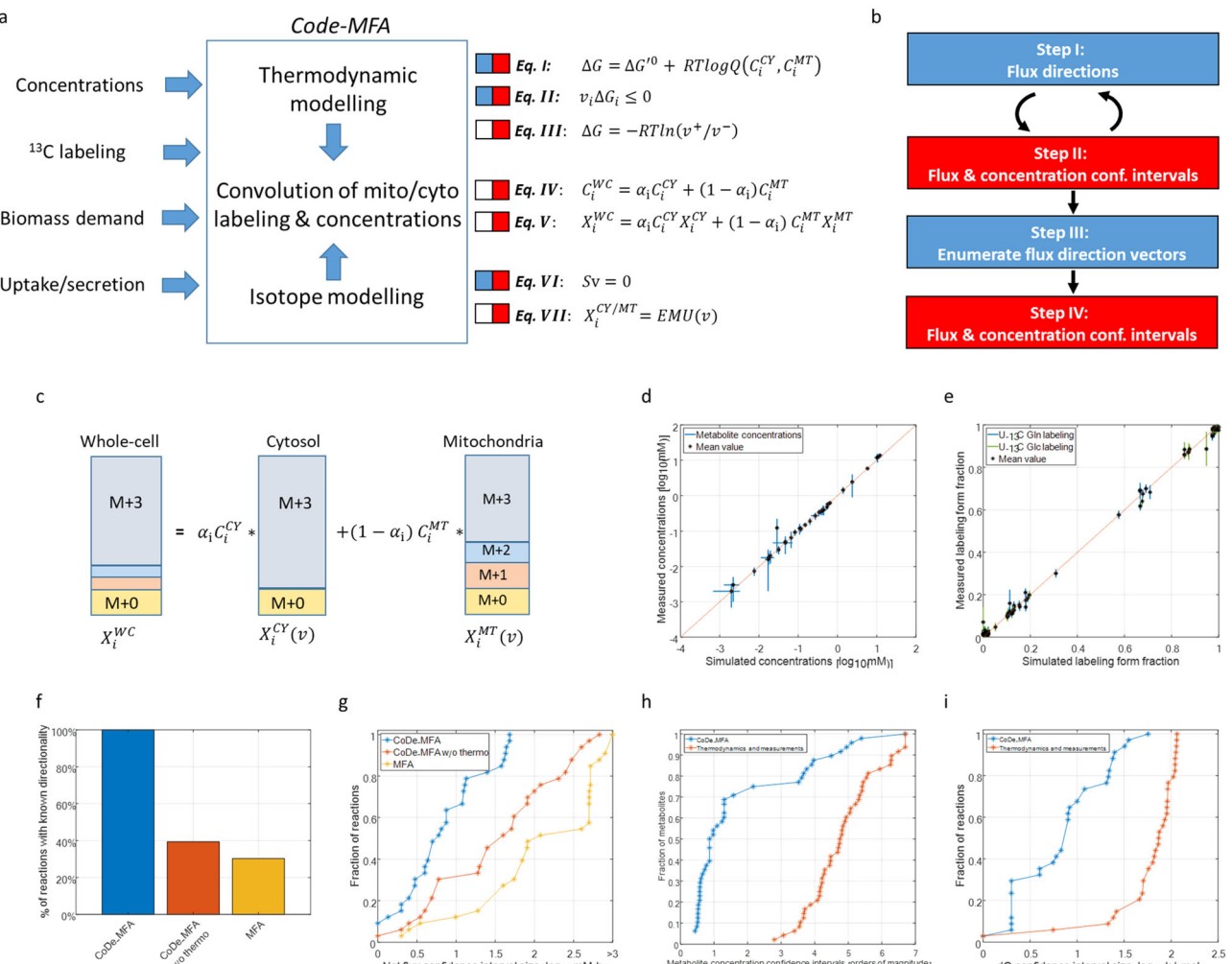

**Fig. 2 | CODE-MFA method and performance inferring compartmentalized fluxes, Gibbs energies, and concentration in HeLa cells. a** CODE-MFA optimization, inferring mitochondrial and cytosolic fluxes, Gibbs free energies, and metabolite concentrations by combining isotope modeling, thermodynamic modeling, and computational deconvolution. Equation (I): reaction Gibbs free energy; $\Delta G$ Gibbs energy, $\Delta G'^0$ standard Gibbs energy, R gas constant, T temperature, Q reaction quotient. Equation (II): second law of thermodynamics. Equation (III): flux-force relationship; $v^+$ and $v^-$ - flux in forward and reverse direction, respectively. Equation (IV): concentration deconvolution; $C_i^{CY}$, $C_i^{MT}$, $C_i^{WC}$ CiWC - simulated concentration of metabolite i in cytosol, mitochondria, and total cellular. $\alpha_i$ - relative cellular volume of cytosol versus mitochondria. Equation (V): Isotopic labeling deconvolution; $X_i^{CY}$, $X_i^{MT}$, $X_i^{WC}$ - simulated isotopic labeling vector of metabolite i in cytosol, mitochondria, and total cellular (determined based on fluxes v). Equation (VI): mass-balance constraint; S stoichiometric matrix, v flux vector. Equation (VII): simulated isotopic labeling vector of metabolite i in cytosol and mitochondria, computed via the Elementary Metabolite Unit (EMU)[39]. Colored bars (blue/red) represent constraints include in each optimization step of the algorithm. **b** CODE-MFA algorithm steps (colors represent the constraints accounted for within each step; according to (**a**). **c** An example isotopic labeling deconvolution equation for a metabolite i having three carbons, with depicted mass-isotopomer distributions for the cytosolic, mitochondrial, and total cellular pools ($X_i^{CY}$, $X_i^{MT}$, $X_i^{WC}$, respectively). **d** The fit been simulated total cellular metabolite concentration (i.e., convolution of simulated mitochondrial and cytosolic concentrations; x axis) and measurements (y axis; data are presented as mean values ± SD, n = 3 independent biological replicates). **e** The fit been simulated total cellular metabolite isotopic labeling (x axis) and experimental measurements (y axis; data are presented as mean values ± SD, n = 3 independent biological replicates). **f** Percentage of reactions in the model whose direction of net flux is inferred by CODE-MFA versus with MFA and CODE-MFA without thermodynamic considerations. **g** Cumulative distribution of reaction net flux confidence interval sizes inferred by CODE-MFA versus MFA and CODE-MFA without thermodynamic considerations. **h** Cumulative distribution of reaction metabolite concentration confidence interval sizes inferred by CODE-MFA versus with strictly thermodynamic analysis. **i** Cumulative distribution of reaction Gibbs energy confidence interval sizes inferred by CODE-MFA versus with strictly thermodynamic analysis.

the total cellular isotopic labeling pattern of a metabolite is more similar to the labeling pattern in the compartment in which its pool size is larger. Hence, the deconvolution of the total cellular isotopic labeling patterns jointly constrains the simulated mitochondrial and cytosolic concentrations and fluxes (determining the compartmentalized isotopic labeling patterns).

We derived the following algorithm to solve the above optimization problem, overcoming the complication due to several non-linear constraints (Fig. 2a, Eq. (II), (III), and (V); Fig. 2c; "Methods"): We utilize Mixed-Integer Linear Programming (MILP) to determine the direction

of net flux through reactions solely based on mass-balance constraints and the second law of thermodynamics (as well physiological measurements; Step I)[40]. Given the inferred flux direction through a subset of the reactions, we formulate a nonconvex optimization problem with linear constraints that combines isotope modeling as well as thermodynamic constraints for these specific reactions (Step II). Sequential dynamic programming (SQP) is used to efficiently solve this problem, computing 95% confidence intervals for all fluxes and metabolite concentrations. The inferred flux and concentration confidence intervals are used as input to iteratively run Step I and II, aiming to infer

the direction of net flux through additional reactions. In case the iterative process converges before the direction of all reactions is uniquely determined, we utilize the derived flux and concentration confidence intervals as a basis to enumerate all thermodynamically possible reaction directionality vectors for the remaining reactions in Step III. Finally, in Step IV, we compute flux and concentration confidence intervals by considering all possible reaction directionality vectors. Overall, the number of reactions for which the direction of net flux is uniquely determined gradually increases throughout the CODE-MFA run (Supplementary Fig. S1), tightening the flux, concentration, and Gibbs energy confidence intervals.

## CODE-MFA infers compartmentalized metabolism in central carbon metabolism in HeLa cells

We applied CODE-MFA to infer central mitochondrial and cytosolic metabolism in cultured HeLa cells. We considered a compartmentalized metabolic network model of central energy metabolism, consisting of 63 reactions, out of which 12 are reactions catalyzed by distinct isozymes in mitochondria and cytosol; and 35 metabolites, out of which 20 are localized in both mitochondria and cytosol (based on the presence of corresponding enzymes in these compartments; "Methods"). We utilized a variety of LC-MS methods to accurately measure the total cellular absolute concentration of 30 metabolites; out of which, we measured steady-state isotopic labeling of 19 central carbon metabolites, when feeding [U-$^{13}$C]-glucose and [U-$^{13}$C]-glutamine ("Methods"; Supplementary Tables S2 and S7). CODE-MFA gradually determined the direction of net flux through up to 61 reactions within 8 iterations before convergence (Steps I–II; Supplementary Fig. S1b). The direction of net flux of the remaining reactions as well as 95% confidence intervals for all fluxes and concentrations in the model were inferred in Steps III and IV (Supplementary Tables S8 and S10). The estimated fluxes and concentrations provided a good match with the measured concentrations and isotopic labeling patterns: for all metabolites, convolution of the estimated mitochondrial and cytosolic concentration is within less than one standard deviation of the measured cellular concentration (Fig. 2d and Supplementary Table S7). For 90% of the measured metabolite isotopic labeling forms, the match between the convoluted compartmentalized estimates and the cellular measurements is less than two standard deviations (Fig. 2e and Supplementary Table S2).

The 95% confidence interval for mitochondria and cytosol fluxes inferred by CODE-MFA markedly varies between reactions (Figs. 2g and 3a–c); for more than 65% of the reactions, the size of flux confidence interval <10 mM/h, including TCA cycle reactions, anaplerotic and cataplerotic reactions, and mitochondrial transporters. As a benchmark, we evaluated the performance of MFA applied solely based on the isotopic labeling of metabolites that are strictly synthesized in mitochondria or in cytosol, whose measured isotopic labeling pattern reflects that of precursor metabolites in a specific compartment; e.g., fatty acid labeling reflects cytosolic acetyl-CoA labeling[41,42], and the isotopic labeling of pyrimidines reflect that of cytosolic aspartate, etc. (Supplementary Tables S2 and S6). While CODE-MFA inferred the direction of net flux through all reactions in the model, MFA inferred the direction of only 30% of the reactions in the model (and through none of the reactions catalyzed by distinct isozymes in the both compartments; Fig. 2f). Furthermore, the median flux confidence interval inferred by MFA was significantly (~20-fold) larger than with CODE-MFA (Wilcoxon P value < 10$^{-7}$; Fig. 2g). As a further benchmark, we implemented a variant of MFA, which similarly to CODE-MFA, utilizes the isotopic labeling patterns of all measured metabolites as input and deconvolutes them into compartment-specific patterns, though without accounting for thermodynamics. This approach also performed markedly worse than CODE-MFA, inferring the direction of net flux through only 40% of the reactions (Fig. 2f), and with a significantly (~threefold) larger flux confidence

interval compared to CODE-MFA (Wilcoxon P value < 10$^{-3}$; Fig. 2g). We compared the inferred compartmentalized fluxes with enzyme mRNA expression in HeLa cells, considering the expression of the specific mitochondrial or cytosolic isozyme for each reaction; utilizing gene expression data from Cancer Cell Line Encyclopedia; CCLE[43]). We found a significant Spearman correlation of 0.72 between the most likely compartmentalized fluxes by CODE-MFA and gene expression levels (Spearman P value = 5 × 10$^{-4}$; Supplementary Fig. S4a); while MFA-derived fluxes are not significantly correlated with gene expression (Spearman P value = 0.48; Supplementary Fig. S4b). Similarly, CODE-MFA-derived fluxes show a significant correlation of 0.54 with enzyme concentration measurements[44] (Spearman P value < 0.03; Supplementary Fig. S4c); while no significant correlation is obtained with MFA fluxes (Spearman P value = 0.36; Supplementary Fig. S4d).

Intracellular metabolite concentrations span ~7 orders of magnitude, ranging from ~10 nM to 100 mM. For 54% of the metabolites in the model having distinct pools in mitochondria and in the cytosol, CODE-MFA inferred their mitochondrial and cytosolic concentration with a 95% confidence interval of less than 1 order of magnitude (Fig. 2h and Supplementary Table S8). For 6 out of 15 metabolites in the employed metabolic network model that are localized in both mitochondria and cytosol, CODE-MFA found a significant difference in their concentration between the two compartments (i.e., no overlap in the 95% confidence intervals; Fig. 3d and Supplementary Table S8). For example, pyruvate and oxaloacetate were found to have more than 100-fold higher concentration in cytosol; and glutamate and citrate more than tenfold higher concentration in mitochondria (Fig. 3d). As a benchmark for inferring compartmentalized concentrations by CODE-MDA, we performed strictly thermodynamic analysis of the measured metabolite concentrations based on the second law of thermodynamics (utilizing Thermodynamic Metabolic Flux Analysis; TMFA)[45,46]. The median concentration confidence interval size inferred by TMFA was ~4 orders of magnitude larger than with CODE-MFA (Wilcoxon P value < 10$^{-11}$; Fig. 2h). Furthermore, CODE-MFA estimations of Gibbs free energy for mitochondrial and cytosolic reactions had a median confidence interval size of 7 kJ/mol (Figs. 2i and 3a–c); significantly (tenfold) lower than with TMFA (Wilcoxon P value < 10$^{-9}$).

CODE-MFA further reduced the uncertainty regarding the mitochondrial and cytosolic concentration of NAD(P)/H, compared to the simplified analytical approach described above (Fig. 1). Specifically, we find that the mitochondrial NADP+/NADPH ratio is at least 3 order of magnitude lower than the cytosolic ratio; with the concentration of NADP+ being markedly lower in mitochondria than in cytosol. The estimated mitochondrial and cytosolic metabolite concentrations and reactions Gibbs energy enabled to revisit the assumptions made by previous estimations of compartmentalized redox cofactor ratios (Fig. 1c). Specifically, CODE-MFA determines that the Gibbs free energy of cytosolic ME1 <−7.6 kJ/mol, explaining the marked (>twofold) underestimation of the cytosolic NADP+/NADPH by previous studies that assumed chemical equilibrium[13,14,16]. Furthermore, CODE-MFA finds that the cytosolic concentration of citrate is more than 2 orders of magnitude lower than the total cellular concentration, explaining the (>twofold) underestimation of the cytosolic NADP+/NADPH ratio, based on total cellar reactant concentration in IDH1[16].

For several reactions we find that different isozymes catalyze flux in opposite direction in mitochondria and cytosol (considering the 95% confidence intervals; Figs. 3a–c and 4a):

Our analysis shows that NADPH-dependent isocitrate dehydrogenases catalyze reductive flux in cytosol (IDH1) and in mitochondria (IDH2), while the NADH-dependent enzyme in mitochondria (IDH3) catalyzes oxidative flux. Oxidative decarboxylation of citrate in mitochondria through IDH3 is highly thermodynamically favorable ($\Delta_r G' <$ −20 kJ/mol), driven by ~100-fold higher concentration of citrate

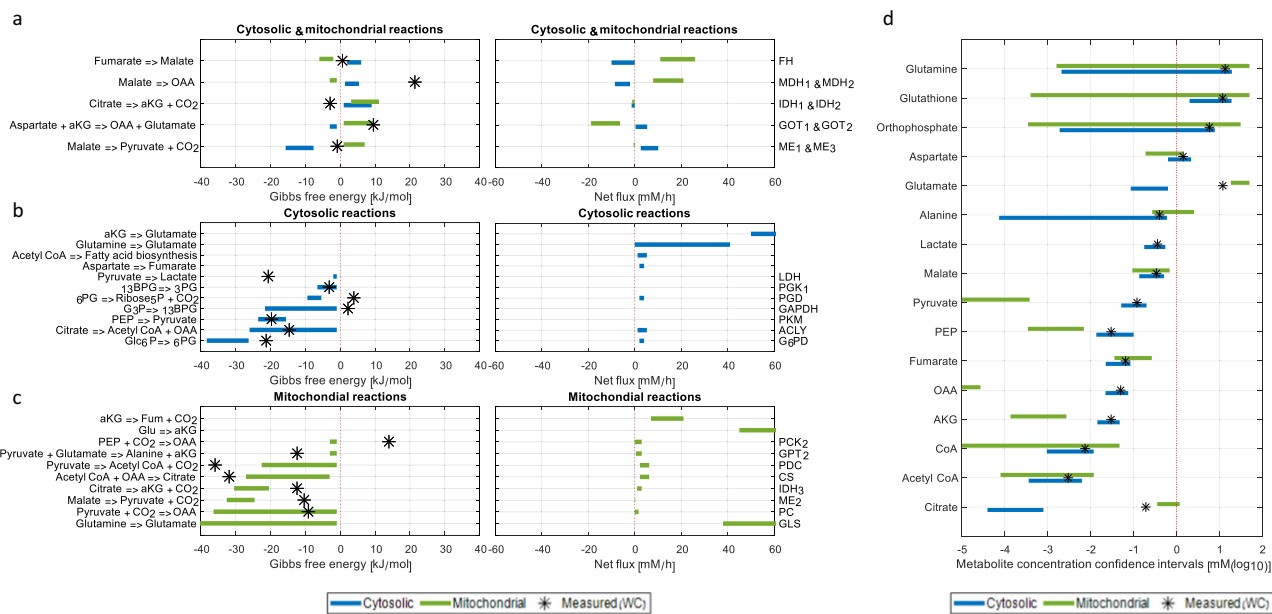

**Fig. 3 | CODE-MFA-derived compartmentalized fluxes, Gibbs energies, and concentration confidence intervals in HeLa cells.** CODE-MFA-derived Gibbs free energies and net fluxes for cytosolic and mitochondrial isozyme (**a**), for strictly cytosolic reactions (**b**), and for strictly mitochondrial reactions (**c**). Blue and green bars represent cytosolic and mitochondrial fluxes and Gibbs energies, respectively; asterisks represent Gibbs energies computed directly based on measured cellular metabolite concentrations. **d** CODE-MFA-derived cytosolic and mitochondrial metabolite concentrations (blue and green bars) and measured cellular concentrations (asterisk).

than α-ketoglutarate in mitochondria (Fig. 3d). Reductive carboxylation of α-ketoglutarate in cytosol is driven by higher α-ketoglutarate than citrate concentration in cytosol; cytosolic α-ketoglutarate is found to be ~tenfold higher in cytosol than in mitochondria, while citrate concentration in cytosol is ~500-fold lower than in mitochondria (Fig. 3d). While the mitochondrial citrate/α-ketoglutarate ratio supports oxidative isocitrate dehydrogenase flux, the mitochondrial, NADPH-dependent IDH2 catalyze reductive flux; facilitated by low NADP+/NADPH ratio (<0.001) in mitochondria (Fig. 1c). Notably, calculating the Gibbs free energy of IDH1/2 based on total cellular concentration of reactants shows that the reaction is thermodynamically favorable in the oxidative direction, emphasizing the need for a compartmentalized view of metabolite concentrations (Fig. 3a). Considering that the relative contribution of reductive IDH1 flux to producing cytosolic citrate is higher than that of IDH2 in mitochondria (as mitochondrial citrate is predominantly produced by citrate synthase), CODE-MFA determines different isotopic labeling patterns of cytosolic and mitochondrial citrate (Fig. 4b). Specifically, CODE-MDA suggests that feeding HeLa cells with [U-13C]-glutamine, 28% of cytosolic citrate will be in the M + 5 form, while only 15% of the mitochondrial citrate is M + 5. The larger pool size of citrate in mitochondria (Fig. 3d) biases the total cellular isotopic labeling towards that in mitochondria, and accordingly, the measured total cellular M + 5 citrate is 15%. Notably, the higher fractional labeling of M + 5 citrate in the cytosol than in mitochondria is directly evident by the high fractional labeling of fatty acids when feeding [U-13C]-glutamine. Specifically, considering that palmitate is strictly synthesized in cytosol, deconvolution of palmitate isotopic labeling[41,42] suggest that ~30% of the cytosolic acetyl-CoA is indeed in the M + 2 form (Supplementary Table S6). Our results regarding the major contribution of reductive IDH1 flux for synthesizing cytosolic citrate in HeLa cells is consistent with our previous report, measuring the isotopic labeling kinetics of citrate in mitochondrial and cytosolic enriched subcellular fractions[6].

CODE-MFA analysis further shows that mitochondrial and cytosolic malate dehydrogenase isozymes (MDH1/2) and glutamate-oxaloacetate transaminase isozymes (GOT1/2) catalyze flux in an opposite direction in each compartment (Figs. 4a and 3a); in accordance with the known activity of the malate-aspartate shuttle, transferring reducing equivalents in the form of NADH from cytosol to mitochondria. Considering that malate dehydrogenase is highly thermodynamically favorable in the reductive direction ($\Delta_r G^{0\prime} = -27$ kJ/mol), driving mitochondrial MDH2 in the oxidative direction (to maintain malate-aspartate shuttle flux) requires a high substrate-to-product concentration ratio. Accordingly, the model suggests ~10,000-fold lower oxaloacetate than malate in mitochondria; with mitochondrial oxaloacetate having a concentration of 100 nM (versus ~100 μM in cytosol; Fig. 3d). Consistent with the low mitochondrial oxaloacetate concentration, mitochondrial phosphoenolpyruvate carboxykinase (PCK2) is predicted to be close to chemical equilibrium ($\Delta_r G\prime > -3$ kJ/mol). Feeding [U-13C]-glutamine, rapid forward–backward flux through PCK2 results in the synthesis of mitochondrial M + 3 phosphoenolpyruvate (PEP; Fig. 4c) and M + 3 OAA (releasing 13CO2 from M + 4 OAA and fixing atmospheric 12CO2): The mitochondrial M + 3 PEP is not observed in total cell measurements, due to the substantially larger PEP pool size in cytosol (Figs. 4c and 3d); while M + 3 OAA condenses with non-labeled acetyl-CoA via citrate synthase, producing the experimentally observed M + 3 citrate (Fig. 4b). The M + 3 OAA further goes through the malate-aspartate shuttle, synthesizing mitochondrial M + 3 aspartate, cytosolic M + 3 OAA, and then cytosolic M + 3 malate. This leads to higher fractional labeling of M + 3 malate in the cytosol (21%) than in mitochondria (10%; Fig. 4d), while the convoluted labeling of M + 3 malate (19%) is biased toward that in cytosol (where malate pool size is large), in agreement with the experimental measurement (Fig. 3d).

## Applying CODE-MFA to quantify mitochondrial and cytosolic metabolism across cell lines

We repeated the analysis to quantify mitochondrial and cytosolic metabolism across a series of cancer cell lines with different tissue of origin, HCT116 (colon cancer), A549 (lung cancer), and LN229 (glioblastoma), performing isotope tracing and absolute metabolite

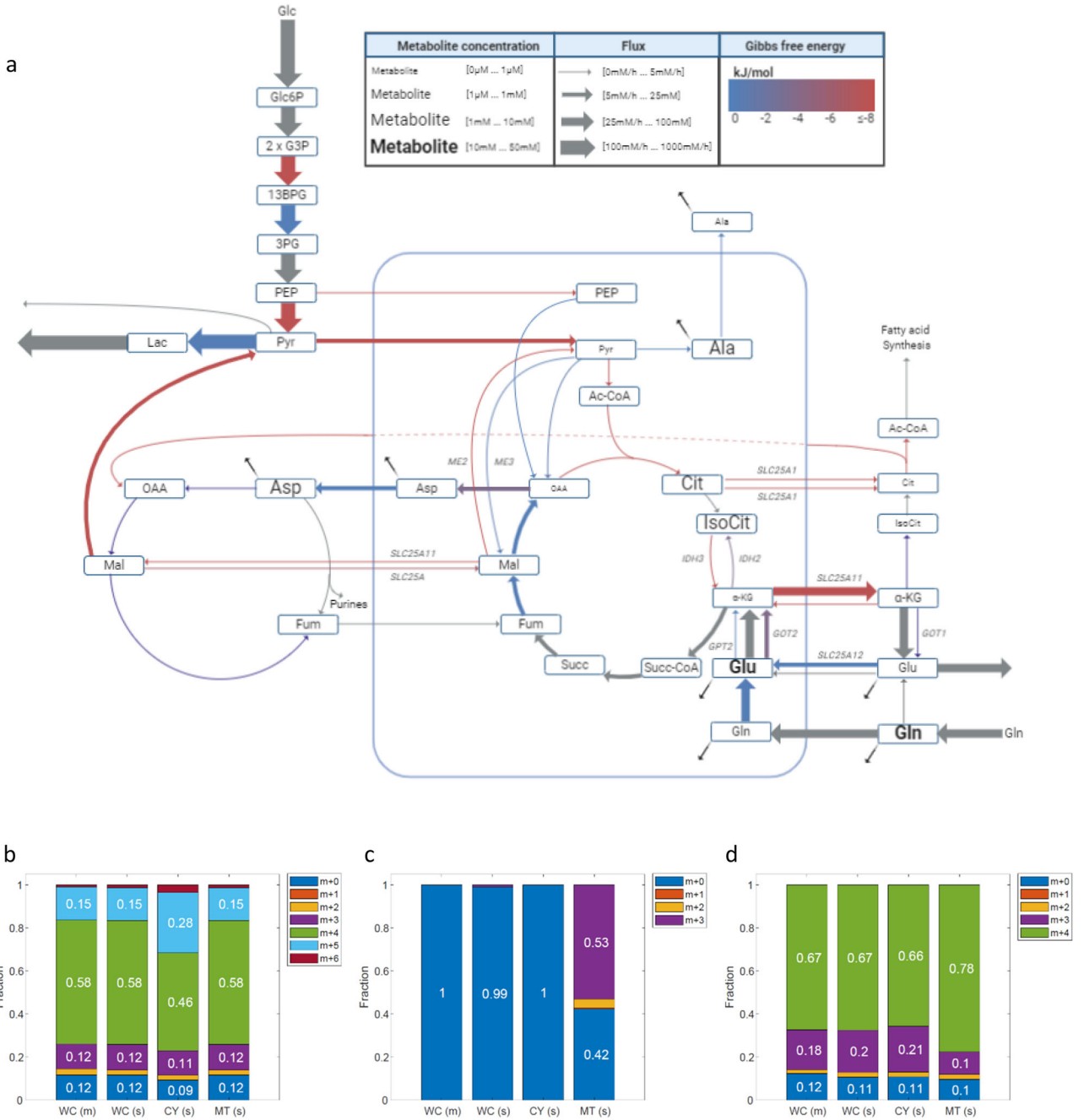

**Fig. 4 | CODE-MFA-derived compartmentalized metabolic activities in HeLa cells and simulated isotopic labeling patterns. a** Inferred fluxes, Gibbs energies, and concentrations. Net fluxes are represented by the arrow width; metabolite concentrations represented by font size; and Gibbs energies by color (see legend). **b–d** Simulated mitochondrial (MT(s)), cytosolic (CY(s)), total (convoluted; WC(S)), and measured (WC(m)) isotopic labeling of citrate (**b**), phosphoenolpyruvate (**c**), and malate (**d**).

concentration measurements in each cell line and applying CODE-MFA (Supplementary Tables S3–S5). As with HeLa cells, the most likely mitochondrial and cytosolic fluxes and concentrations inferred by CODE-MFA provided a good match with the measured concentrations and isotopic labeling patterns in all cell lines (Supplementary Table S7 and Supplementary Fig. S5). For all cell lines, the performance of CODE-MFA outperformed that of standard MFA: CODE-MFA determined the direction of net flux through more than 85% of the reactions in the model for all cell lines, while standard MFA inferred net flux direction trough no more than 31% of the reactions (Supplementary Fig. S6). The median flux confidence interval inferred by CODE-MFA is more than one order of magnitude smaller than with MFA in all cell

lines (Wilcoxon p value per cell line <10⁻⁵; Supplementary Fig. S6). The median metabolite concentration confidence interval size inferred by CODE-MFA is more than 3 orders of magnitude smaller than with strictly thermodynamic analysis in all cell lines (Wilcoxon $P$ value < 10⁻⁹; Supplementary Fig. S7). The most likely compartmentalized fluxes inferred by CODE-MFA are significantly correlated with gene expression levels of the corresponding enzymes for all cell lines (Spearman $P$ value < 0.02; Supplementary Fig. S8).

The resulting compartmentalized flux programs are overall highly similar, with a median pairwise Pearson correlation of 0.84 (all pairwise Pearson $P$ value < 4 × 10⁻⁴; Fig. 5). A specific metabolic system showing major variability between cell lines is malate-aspartate shuttle. For

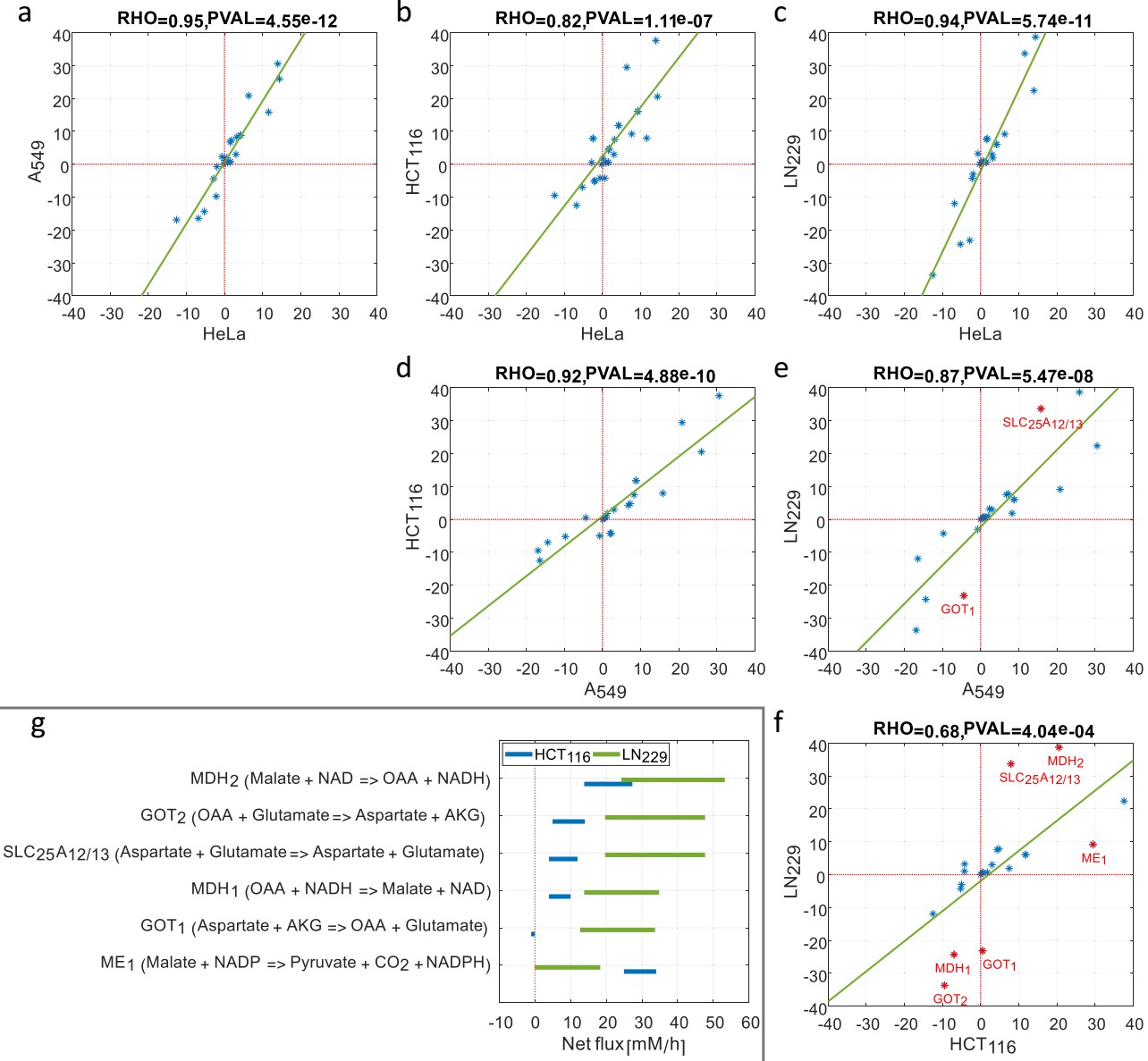

**Fig. 5 | Correlation of CODE-MFA-derived fluxes for HeLa, HCT116, A549, and LN229 cell lines. a** Two-sided Pearson correlation of CODE-MFA-derived fluxes for HeLa and A549. **b** Two-sided Pearson correlation of CODE-MFA-derived fluxes for HeLa and HCT116. **c** Two-sided Pearson correlation of CODE-MFA-derived fluxes for HeLa and LN229. **d** Two-sided Pearson correlation of CODE-MFA-derived fluxes for A549 and HCT116. **e** Two-sided Pearson correlation of CODE-MFA-derived fluxes for A549 and LN229. **f** Pearson correlation of CODE-MFA-derived fluxes for HCT116 and LN229. **g** CODE-MFA-derived net fluxes for fluxes with large difference per (**f**) (HCT116 vs. LN229).

example, several mitochondrial and cytosolic reactions in this shuttle have significantly lower flux in HCT116 versus in LN229 (Fig. 5f, g); including both cytosolic and mitochondrial malate dehydrogenase (MDH1 and MDH2, respectively), glutamate-oxaloacetate transaminase (GOT1 and GOT2, respectively), and the mitochondrial aspartate transporter (SLC25A12/13). While having lower malate-aspartate shuttle flux than LN229 (and hence low net transport of NADH from the cytosol to mitochondria), cytosolic NAD + /NADH ratio in HCT116 is maintained through higher cytosolic malic enzyme (ME) flux-producing pyruvate that is reduced to lactate while oxidizing NADH to NAD+ (Fig. 5g).

To validate the importance of high ME1 flux in HCT116, we found that inducible silencing of ME1 leads to a significant drop in HCT116 cell proliferation (*t* test *P* value < 0.01), while its silencing in LN229 leads to no significant change in growth (Fig. 6b). A major contribution of ME1

flux to support NADPH production in HCT116 is supported by a significant drop in NADPH/NADP ratio upon ME1 silencing; while no significant drop in NADPH/NADP ratio is observed up ME1 silencing in LN229 (Fig. 6a; *t* test *P* value < 0.01). Our finding of a major flux through cytosolic malic enzyme in HCT116 are consistent with a recent report regarding the importance of ME1 in these cells for maintaining redox balance and cell growth[47]. Accordingly, considering that glutamine is a major TCA anaplerotic source, we hypothesized that glutamine removal would significantly harm HCT116 having high malic enzyme flux. Indeed, glutamine removal led to >50% drop in cell number after 24 h; a significantly larger drop than in a control cell line, LN229 (in which we identify lower ME1 flux; Fig. 6d; *t* test *P* value < 0.03). Consistently, we detect a larger drop in the concentration of TCA cycle metabolites upon glutamine removal in HCT116 (Fig. 6c; *t* test *P* value < 0.01).

a
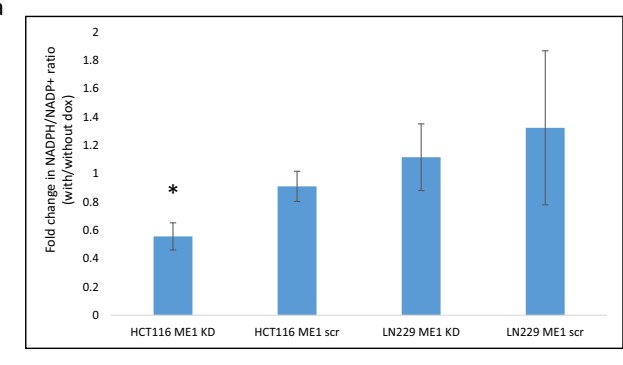

b
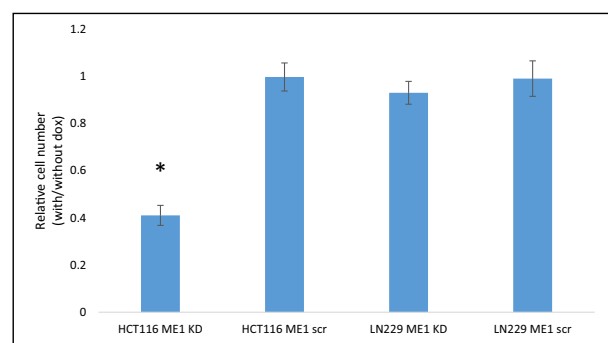

c
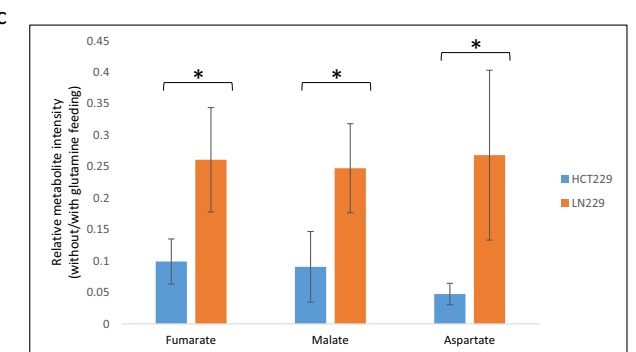

d
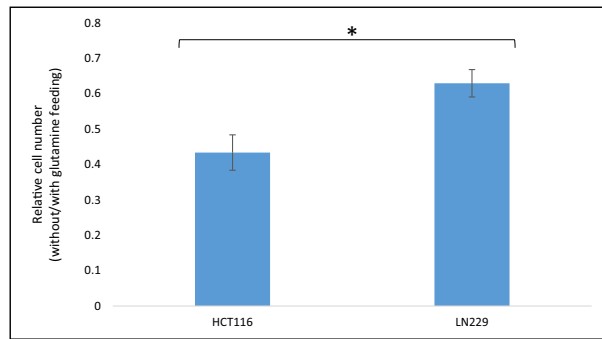

**Fig. 6 | Validate the importance of high ME1 flux in HCT116 vs. LN229 cell lines.**
**a** NADPH/NADP+ ratio in doxycycline-inducible ME1 knockdown (KD) and ME1 control (scr) in HCT116 and LN229 cells with/without dox treatment (*$P < 0.01$ by two-sided $t$ test; data are presented as mean values $\pm SD$, $n = 3$ independent biological replicates). **b** Growth (5 days) of doxycycline-inducible ME1 knockdown (KD) and ME1 control (scr) in HCT116 and LN229 cells with/without dox treatment (*$P < 0.01$ by two-sided $t$ test; data are presented as mean values $\pm SD$, $n = 3$

independent biological replicates). **c** Intracellular relative metabolites intensities in HCT116 and LN229 cells without/with glutamine feeding (*$P < 0.01$ by two-sided $t$ test; data are presented as mean values $\pm SD$, $n = 3$ independent biological replicates). **d** Relative cell number, 24-h without/with glutamine feeding, of HCT116 and LN229 cells (*$P < 0.03$ by two-sided $t$ test; data are presented as mean values $\pm SD$, $n = 3$ independent biological replicates).

## Discussion

We presented a generic approach that enables for the first time to infer mitochondrial and cytosolic fluxes and metabolite concentrations via measurements performed with intact cells under physiological conditions (without requiring subcellular fractionation, potentially perturbing metabolic activities). This was made possible by combining flux and thermodynamic modeling with deconvolution of cellular metabolic measurements into their mitochondrial and cytosolic counterparts. Previous approaches demonstrated the potential of deconvoluting metabolite concentration measurements into compartmentalized pool sizes[36,48]. Here we show how metabolite isotopic labeling patterns measured in isotope tracing experiments can be deconvoluted into distinct mitochondrial and cytosolic counterparts and facilitate direct inference of compartmentalized fluxes. While integrative compartmentalized modeling of isotope tracing and reaction thermodynamics is computationally hard[37,48], we devise an optimization-based approach for inferring compartment-specific fluxes and concentrations corresponding confidence intervals. While focusing on cytosolic and mitochondrial metabolism, our analysis is not biased by metabolic activities in nuclei due to free diffusion of small molecules though NPCs into the cytosol[49]; hence, the inferred cytosolic fluxes and metabolite concentrations represents averaged values in cytosol and nucleus. Notably, compartmentalized metabolic activities in other organelles such as ER, Golgi apparatus, peroxisomes, and lysosomes could potentially bias some of the inferred concentrations and fluxes in mitochondria and cytosol. While our method is based on measurements of metabolite labeling under isotopic steady state, it can be further extended to model metabolite isotopic labeling kinetics (in accordance with non-stationary MFA)[50]. This could potentially

tighten estimated flux and concentration confidence intervals as isotopic labeling kinetics depend on metabolite concentrations; further linking and constraining the simulated concentrations and isotopic labeling dynamics. Further improvement of mitochondrial and cytosolic flux and concentration estimates could potentially be achieved by incorporating measurements of the isotopic labeling of collisional fragments of metabolites via MS/MS[51–53]. Overall, applied to a series of proliferating cancer cell lines, our method is shown to detect variation in mitochondrial and cytosolic redox cofactor metabolism across cell lines that are non-observable with current flux inference techniques.

Recent studies suggest a variety of mitochondrial and cytosolic isozymes promoting cancer-specific proliferation, highlighting a growing need for methodologies for probing subcellular level metabolic activities: Oncogenic mutations in either NADP-dependent cytosolic and mitochondrial isocitrate dehydrogenases (IDH1 and IDH2, respectively) in tumors result in distinct metabolic reprogramming[54,55]. Recently, major variability in cancer cell reliance on cytosolic (SHMT1) versus mitochondrial (SHMT2) serine-derived one-carbon metabolic flux was shown across tumors, suggesting the reduced folate carrier (RFC) as a marker for tumor sensitivity for drug targeting of the cytosolic pathway[56]. In accordance with our results here regarding high cytosolic NADPH-dependent malic enzyme (ME1) flux in HCT116, ME1 was reported to be a major producer of cytosolic NADPH in some cancers[57], whose silencing suppress cancer cell growth[47,58], and is considered a predictive marker and for prognosis for radiation therapy in cancer[59,60]. We expect that providing means to infer metabolic activities in a subcellular resolution would be instrumental for a variety of studies on metabolic dysfunction in cancer and in other human disease.

## Methods

### Materials

Water and organic solvents were obtained from Merck & Co (LiChrosolv, LC-MS grade, Germany), buffer additives for HPLC and ammonia solution, 25%, formic acid (for LC-MS) and NaOH were purchased from Fluka Analytical Sigma-Aldrich (Germany) and Merck & Co (Germany), respectively. [U-$^{13}$C]-glucose (CLM-1396-1), [U-$^{13}$C]-glutamine (CLM-3612) and [U-$^{13}$C]-lactic acid (CLM-1579) were bought from Cambridge Isotope Laboratories.

### Cell culture

HeLa (CCL-2), HCT116 (CCL-247), A549 (CCL-185) and LN229 (CRL-2611) cells (purchased from ATCC, USA) were cultured in high-glucose Dulbecco's Modified Eagle's Medium (Biological Industries 01-055-1 A, 25 mM glucose) supplemented with 10% (v/v) dialyzed fetal bovine serum (HyClone SH30079.03), 1% penicillin–streptomycin-amphotericin B solution (Biological Industries 03-033-1B), and 4 mM glutamine.

All cell lines were cultured using standard procedures in a 37 °C humidified incubator with 5% $CO_2$. Cell lines were tested for mycoplasma using EZ-PCR mycoplasma detection kit (Biological Industries, 20-700-20).

In total, $2 \times 10^5$ cells from each cell line were seeded in 6-cm plates (three repeats for each cell type) and grown for 24 h before metabolite extraction. Plates were also seeded for cell count and packed cell volume (PCV) measurements for normalization using a coulter counter (Bekman Coulter Counter, USA). For each experiment, cells were washed twice with PBS before being cultured with the indicated media.

### Isotope tracing

For labeling experiments, isotope tracing was performed by feeding cells with either [U-13C]-glucose (25 mM) or [U-13C]-glutamine (4 mM) for 24 h, or with [U-13C]-Lactic acid (50 µM) for 3 h before metabolite extraction. To verify that cells reached metabolic steady state in 24-h incubation, we compared intracellular metabolite concentrations made after 24-h with those measured after 48-h incubation. We found overall comparable concentrations in 24-h and 48-h (Supplementary Fig. S2), with no statistically significant change in the concentration of up to 70% of the metabolites (considering concentrations 95% confidence interval). While metabolite concentrations were found to span ~4 orders of magnitude, the maximal change in concentration of a metabolite between 24-h and 48-h incubation was lower than 40% (Supplementary Fig. S2 and Supplementary Table S13). We confirmed that cells reached isotopic steady state by repeating the [U-13C]-glucose and [U-13C]-glutamine feeding in HeLa cells for 48-h incubation and compared the measured metabolite isotopic labeling patterns to those measured after 24-h incubation (focusing on metabolites metabolite measurements using SeQuant ZIC-pHILIC column). We found a strong match between metabolite isotopic labeling patterns after 24-h and 48-h incubation, with a median difference of less than 1%; and the 95-percentile largest difference lower than 3% (Supplementary Table S14).

### Cell proliferation assay

In all, $1.8 \times 10^5$ cells per well were seeded in 6-cm plates. For the 5-day (120 h) growth assay, media were replaced on day 3. Cell number was counted via a Multisizer Coulter Counter (Beckman Coulter) (Fig. 6b).

In total, $0.6 \times 10^6$ cells per well were seeded in 6-cm plates. Sixteen hours later, media were changed to glutamine-free media. 24 h after, cells were counted. Cell number was measured via a Multisizer Coulter Counter (Beckman Coulter) (Fig. 6d).

### Generation of inducible shRNA ME1 knockdown

ShRNA sequences were cloned into lentiviral vector Tet-pLKO-puro (purchased from AddGene, 21915). The shRNA target sequences were as follows:

sh1: 5′-GGGCATATTGCTTCAGTTC-3′
sh2: 5′-GCCTTCAATGAACGGCCTATT-3′
shScr: 5′-AACAAGATGAAGAGCACCAA-3′.

To generate active lentivirus, $4.6 \times 10^6$ 293T cells were cultured in DMEM growth medium supplemented with 10% fetal bovine serum in 100-mm plate, Cells were at 80% confluency before seeding; passage number was lower than 12. After 24 h, cells were transfected using a 4:2:1 ratio of Tet-pLKO-puro: psPAX2 (Addgene catalog number 12260):pMD2.G (Addgene catalog number 12259) using PEI (DNA:PEI = 1:3). After 24 h cell growth, the medium was replaced with fresh DMEM, supernatant-containing viruses was collected 72 h post transfection, and filtered through a 0.45-µm sterile filter.

For cell transduction: $2.5 \times 10^5$ HCT116 cells were seeded in a six-well plate. After 24 h, 200 µl of the viral solution was added to the cells. Cells were transduced with the presence of 10 µg/mL polybrene. After 24 h, the media were replaced with a fresh media. Transduced cells were selected using 3 mg/mL puromycin for 72 h. For shRNA induction, 1 µg/mL doxycycline was used.

### LC-MS analysis

For the measurement of intracellular metabolites, cells were washed with 2 mL of ice-cold PBS twice on ice. The cells were extracted quickly in 300 µL volume of methanol/acetonitrile/water (50:30:20, v/v/v) solution at −20 °C on dry ice by scraping. For the measurement of media metabolites, 50 µL of media was mixed with 200 µL volume of methanol/acetonitrile (75:25, v/v) ice-cold solution.

For lipids saponification, the following protocol was used: cells were extracted quickly in 1 mL of PBS by scraping and then spanned at 500 ×$g$ for 5 min at 22 °C. 200 µL volume of ethanol/water (80:20, v/v) solution (0.02 M NaOH) was added to pellets and then transferred to glass HPLC vials to incubate in 66 °C for 1 h. Then samples were mixed with 100 µL of acetonitrile (5% formic acid).

For uptake/secretion experiments, unlabeled internal standard mixes were used. Measurements were performed with a method of standards additions. Extracted media samples were spiked with appropriate standards mix (1:1, v/v). Concentrations of standards were determined for each cell line based on their specific consumption rates.

For NAD(P)(H) measurements, the following protocol was used[61]: cells were seeded on 6 cm plates in the non-labeled media. About 16 h later, media were changed to new non-labeled media. 24 h later media were aspirated and without any washing, 150 µl special extraction solution was added to each plate (40 ACN:40 MeOH:20 H2O with 0.1 M Formic acid), plated were covered and left on ice for 3 min, cells were detached form the plate surface by a plastic cell scraper. The content of the plate including cell pellet was transferred to 1.5-mL micro centrifuge tube. In all, 12 µl ammonium bicarbonate was added to each tube. The tubes were briefly vortexed and then frozen at −80 °C for 30 min−2 h. After thawing, tubes were vortex again. Tubes were centrifuged at 20,000× $g$ for 20 min at 4 °C to remove pellet and stored at −80 °C until the analysis. Samples were centrifuged again with the same parameters directly before transferring them to LC-MS vials (Fig. 6a). All the metabolite samples were stored at −80 °C for at least 2 h. Protein-free metabolite extractions were prepared by spinning the samples at 20,000× $g$ for 20 min at 0 °C twice. Samples were subsequently analyzed using liquid chromatography-mass spectrometry (LC-MS) method.

Cellular and media extracts were chromatographically separated on a SeQuant ZIC-pHILIC column ($2.1 \times 150$ mm, 5 µm, EMD Millipore) with an HPLC system (Ultimate 3000 Dionex LC system, Thermo Fisher Scientific, Inc., USA). Flow rate was set to 0.2 mL min$^{-1}$, column compartment was set to 30 °C, and autosampler tray maintained 4 °C. Mobile phase A consisted of 20 mM ammonium carbonate and 0.01% (v/v) ammonium hydroxide. Mobile Phase B was 100% acetonitrile. The

mobile phase linear gradient (%B) was as follows: 0 min 80%, 12–15 min 20%, 15.1 min 80%, 23.0 min 80%.

A chromatographic column SeQuant ZIC-HILIC (4.6 × 150 mm, 5 μm, EMD Millipore) were used for determination of fumaric acid, alanine and asparagine. Separation was performed with flow rate of 1.0 mL min⁻¹, column compartment was set to 50 °C, and autosampler tray maintained 4 °C. Both mobile phases water and acetonitrile were spiked with 0.1% of formic acid. The following mobile phase linear gradient (%B) was used: 0–6 min 95%, 15 min 60%, 20–26 min 20%, 26.1 min 95%, 34 min 95%.

A reverse-phase chromatographic column (2.1 × 100 mm, 1.7 μm, Kinetex C18 EVO, Phenomenex) was utilized for analysis of palmitic acid labeling. A ramp gradient of mobile phase A (5% acetonitrile, 95% water, 10 mM ammonium acetate) and B (50% acetonitrile, 50 isopropanol, 10 mM ammonium acetate) was used with flow rate of 0.4 mL min⁻¹. The mobile phase ramp gradient (%B) was as follows: 0–1.9 min 75%, 2–5 min 100%, 5–8 min 75%.

Mass spectrometry detection was performed using a Q Exactive Hybrid Quadrupole Orbitrap high-resolution mass spectrometer with an electrospray ionization source (ESI, Thermo Fisher Scientific, Inc., USA) for all chromatographic methods. Ionization source parameters were the following: sheath gas 25 units, auxiliary gas 3 units, spray voltage 3.3 and 3.8 kV in negative and positive ionization mode respectively, capillary temperature 325 °C, S-lens RF level 65, auxiliary gas temperature 200 °C. Metabolites were analyzed in the range 72–1080 $m/z$, AGC target was set to $3 \times 10^6$. A single ion monitoring method was used for the analysis of metabolites with low concentration: pyruvic, lactic, oxaloacetic, fumaric and palmitic acids, coenzyme A and acetyl coenzyme A, alanine, and asparagine. The width of ion range for each compound was defined individually in order to collect information for all isotopic forms within a single range. AGC target was set to $1 \times 10^6$ for SIM measurements. An additional concentration protocol was implemented for metabolites measured with single ion monitoring mode to increase sensitivity. Specifically, the cell extracts after first centrifugation were evaporated with nitrogen and reconstituted with 50 μL of methanol/acetonitrile/water (50:30:20, v/v/v). After vortex and additional centrifugation at 20,000× g for 20 min at 0 °C samples were subsequently analyzed using liquid chromatography-mass spectrometry (LC-MS) method with increased injection volume of 10 μL. Retention time of metabolites in the chromatogram were identified by corresponding pure chemical standards. Concentration of metabolites were quantified using chemical standards via an isotope-ratio approach, as described previously[62,63] (see Supplementary Data and Supplementary Table S15). Data were analyzed with commercially available software, MAVEN[64]. A CO₂ concentration of 1.2 mM was assumed[6,65], while repeating the CODE-MFA analysis of HeLa cells to confirm the robustness of our results to threefold lower and 3-fold higher CO₂ concentrations (0.4 mM and 3.6 mM). We found that the inferred compartmentalized fluxes were highly similar to those inferred using the original 1.2 mM concentration, with a pairwise Pearson correlation >0.98 (Pearson $P$ value <$10^{-10}$).

## Gene expression analysis

We utilized RNASeq data from the Cancer Cell Line Encyclopedia (CCLE)[43] to identify enzymes with especially low expression that were assumed nonfunctional in specific analyzed cell lines (considering a common and stringent RPKM cutoff of 0.5): In all cell lines analyzed here, PCK1 (phosphoenolpyruvate carboxykinase 1; catalyzing cytosolic synthesis of phosphoenolpyruvate from oxaloacetate) and GPT (glutamic-pyruvate transaminase; catalyzing cytosolic transamination of pyruvate to alanine) were found to be nonfunctional. ME3 (mitochondrial NADP-dependent malic enzyme 3) was found to be nonfunctional in HCT116 and LN229 cell lines (Supplementary Fig. S3).

## Inferring mitochondrial and cytosolic redox cofactor ratios

We infer the cytosolic NADP+/NADPH ratio based on thermodynamics analysis of 6PGD in cytosol (according to Eqs. (1–3)). To compute the 95% confidence interval for the derived NADP+/NADPH ratio, we randomly sampled values for the concentrations of R5P, 6PG and for the fractional isotopic labeling $R5P_{m+5}$ and $6PG_{m+5}$ from a normal distribution whose mean and standard deviation are determined based on the experimentally measured values (considering biological replicates); choosing NADP+/NADPH ratios between 2.5th and 97.5th percentiles (denoted $r^{lb}$ and $r^{ub}$, respectively).

The inferred cytosolic NADP+/NADPH ratio was used to compute the range of possible concentrations NADP+ and NADPH in mitochondria and in cytosol using Linear Programming (using Eqs. (4–5)):

$$\min/\max$$
$$NADP_{CY}, NADP_{MT}, NADPH_{CY}, NADPH_{MT} \qquad (6)$$

$$s.t.$$

$$r^{lb} NADPH_{CY} \leq NADP_{CY} \leq r^{ub} NADPH_{CY} \qquad (6a)$$

$$NADP_{WC}^{lb} \leq (1-\alpha)^* NADP_{MT} + \alpha^* NADP_{CY} \leq NADP_{WC}^{ub} \qquad (6b)$$

$$NADPH_{WC}^{lb} \leq (1-\alpha)^* NADPH_{MT} + \alpha^* NADPH_{CY} \leq NADPH_{WC}^{ub} \qquad (6c)$$

$$NADP_{MT}, NADP_{CY}, NADPH_{MT}, NADPH_{CY} \geq 0 \qquad (6d)$$

where Eq. (6a) enforces the NADP+/NADPH ratio within the confidence intervals estimated above; and Eq. (6b, c) constrain the convoluted concentration of NADP+ and NADPH to be equal to the measured values (where $NADP(H)_{WC}^{lb}$ and $NADP(H)_{WC}^{ub}$ represent the measured concentration of NADP(H) minus and plus two standard deviations, respectively). The relative volume of cytosol out of total cellular volume is denoted $\alpha$. To compute the range of possible mitochondrial NADP+/NADPH ratios, we constrain this ratio to a series of increasing values (from $10^{-5}$ to $10^5$), considering the minimal and maximal ratios in which a feasible solution was found.

We infer the cytosolic NAD+/NADH ratio based on thermodynamics analysis of lactate dehydrogenase (LDH) in the cytosol:

$$\frac{NAD+_{CY}}{NADH_{CY}} = \frac{Pyruvate_{CY}}{Lactate_{CY}} * \frac{J^-}{J^+} e^{-\frac{\Delta G_0}{RT}} \qquad (7)$$

As pyruvate is present in both mitochondria and cytosol, we estimated the cytosolic concentration based on the second law of thermodynamics, assuming net transport of cytosolic pyruvate into mitochondria through the mitochondrial pyruvate carrier (MPC) and considering the transporter's standard Gibbs free energy

$$\alpha^* Pyruvate_{CY} + (1-\alpha)^* Pyruvate_{MT} = Pyruvate_{WC} \qquad (8)$$

$$\Delta G = \Delta G_0(MPC) + RT^* \ln \frac{Pyruvate_{MT}}{Pyruvate_{CY}} \qquad (9)$$

The cytosolic NAD+/NADH ratio was used to compute the compartmentalized concentration of NAD+ and NADH and mitochondrial ratio as described above for NADP/NADPH+.

## Gibbs free energy of mitochondrial transporters

The standard Gibbs free energy of mitochondrial transporters was calculated by considering the difference in pH between cytosol and

mitochondria and membrane potential, as previously described[66]:

$$\Delta_r G_k^{\prime 0} = -N_H RT \ln\left(10^{\Delta pH}\right) - FQ\Delta\varnothing + S_k^T \Delta_f G^{\prime 0} \qquad (10)$$

where $N_H$ is the net number of hydrogen ions transported from initial to final compartment, $R$ is the gas constant, $T$ is the temperature, $\Delta pH$ is the difference in pH between initial and final compartment, F is Faraday's constant, Q is the net number of charges transported from initial to final compartment, $\Delta\varnothing$ is the difference between the electrical potential of the initial and final compartments, $S$ is the stoichiometric matrix, and $\Delta_f G^{\prime 0}$ is the standard Gibbs energy of formation of the species involved in the reaction. For this analysis, we used $\Delta\varnothing$ of 150 mv[66,67], cytosolic pH level of 7 (see ref. 66), and mitochondrial pH level of 7.5 (estimates range between 7.4–8.2 (see refs. 66,68–70)). We evaluated the robustness of our analysis performed for HeLa cells, considering potential mitochondrial pH levels of 8 and 8.5. The estimated compartmentalized fluxes were highly similar to those inferred using the original mitochondrial pH level of 7.5, with a pairwise Pearson correlation >0.98 (Pearson $P$ value < $10^{-10}$).

We calculated the standard Gibbs free energies of 7 mitochondrial transporters according to Eq. (10) (Supplementary Table S11) and considered 30% standard deviation.

## Compartmentalized deconvoluted metabolic flux analysis (CODE-MFA)

CODE-MFA searches for the most likely mitochondrial and cytosolic net fluxes under metabolic steady state (denoted $\upsilon^{net}$), compartmentalized metabolite concentrations (denoted C), and reactions' standard Gibbs free energies ($\Delta G^0$), providing optimal match with concentration and isotopic labeling measurements performed in intact cells, as well as metabolite uptake and secretion rates, and biomass growth requirements. Specifically, it is formulated as the following optimization problem:

$$\min_{C^{SIM}, \upsilon^{net}, \Delta G^0} \sum_{i=1}^{N_c} \left( \frac{C_i^{WC} - \left(\alpha * C_i^{SIM,CY} + (1-\alpha) * C_i^{SIM,MT}\right)}{\sigma_{C_i^{WC}}} \right)^2 + \qquad (11)$$

$$\sum_{t=1}^{K}\sum_{i=1}^{N_t}\sum_{j=1}^{N_i} \left( \frac{X_{t,i,j}^{WC} - \left(\beta_i(C) * X_{t,i,j}^{SIM,CY}\left(C, \upsilon^{net}, \Delta G^{\prime 0}\right) + (1-\beta_i(C)) * X_{t,i,j}^{SIM,MT}\left(C, \upsilon^{net}, \Delta G^{\prime 0}\right)\right)}{\sigma_{X_{t,i,j}^{WC}}} \right)^2$$

$$s.t.$$

$$S\upsilon^{net} = 0 \qquad (11a)$$

$$\upsilon^{net,lb} \leq \upsilon^{net} \leq \upsilon^{net,ub} \qquad (11b)$$

$$\ln[C]_{lb} \leq \ln\left[C^{SIM}\right] \leq \ln[C]_{ub} \qquad (11c)$$

$$\Delta G_{lb}^{\prime 0} \leq \Delta G^{\prime 0} \leq \Delta G_{ub}^{\prime 0} \qquad (11d)$$

$$sign(\upsilon_i^{net}) * [\Delta G^{\prime 0} + RT \ln Q\left(C^{SIM}\right)] \leq 0 \qquad (11e)$$

where, Eq. (11a) enforces stoichiometric mass balance ($S_{i,j}$ representing the stoichiometric coefficient of metabolite $i$ in reaction $j$; see compartmentalized network model and stoichiometric coefficients in Supplementary Table S1). Equation (11b) enforces lower and upper bounds on net fluxes ($\upsilon^{net,lb}$ and $\upsilon^{net,ub}$ respectively),

based on experimentally measured net uptake/secretion rates and estimated cellular demand for cell proliferation (Supplementary Tables S1 and S9). Equation (11c) enforces lower and upper bounds on metabolite concentrations; lower bound is based on minimum physiological concentration, and upper bound is based on total cellular measurements, while considering a metabolite is fully concentrated in one compartment. Equation (11d) enforces lower and upper bounds on the searched metabolite standard Gibbs free energy, considering previous estimates of free energies and their 95% confidence intervals using the group contribution method[33–35]. Equation (11e) constrains the direction of net flux through each reaction based on the sign of Gibbs free energy, in accordance with the 2nd law of thermodynamics[71].

The first term in the objective function minimizes the variance-weighted sum of squared residual of the experimentally measured metabolite concentrations ($C^{WC}$) versus the convoluted simulated concentrations in cytosol ($C^{SIM,CY}$) and mitochondria ($C^{SIM,MT}$), considering the relative volume of cytosol ($\alpha$) and mitochondria ($1-\alpha$). The standard deviation in the measurements of metabolite concentrations is denoted $\sigma_{C^{WC}}$. The relative volume of the two compartments was calculated assuming that cytosol and mitochondria accounts for ~80% and ~20% of total cellular volume, respectively[5]. We evaluated the robustness of our analysis performed for HeLa cells, considering a range of potential mitochondrial volumes (4%, 12%, 20%)[5,72–78]. The estimated compartmentalized fluxes were highly similar to those inferred using the original mitochondrial 20% volume, with a pairwise Pearson correlation >0.95 (Pearson $P$ value < $10^{-10}$). The second term in the objective function minimizes the variance-weighted sum of squared residual of the experimentally measured metabolite labeling patterns versus the convoluted simulated labeling in cytosol and mitochondria. We denote the measured mass-isotopomer distribution (MID) in the $t$th isotope tracing experiment, for the $i$th metabolite under isotopic steady state, by $X_{t,i}^{WC}$. The relative fraction of M + j form with $X_{t,i,j}^{WC}$; i.e., the fraction of the metabolite pool having j labeled carbons; and the standard deviation in this measurement denoted $\sigma_{X_{t,i,j}^{WC}}$. We denote by $K$ (=3) the number of isotope tracing experiments performed (using [U-$^{13}$C]-glucose, [U-$^{13}$C]-glutamine, and [U-$^{13}$C]-lactate); by $N_t$ the number of metabolites whose labeling pattern is measured in the $t^{th}$ isotope tracing experiment; and by $N_i$ the number of isotopic labeling forms in the $i^{th}$ metabolite. The metabolite MIDs in cytosol and mitochondria, denoted $X_{t,i}^{SIM,CY}$ and $X_{t,i}^{SIM,MT}$, respectively, are uniquely determined based on the forward and backward fluxes[39], derived from the net flux, concentration, and Gibbs free energy variables (as shown below). The convolution of the simulated MIDs of the $i$th metabolite in mitochondria and cytosol accounts for the relative pool size of the metabolite in each compartment; the relative pool size in cytosol is denoted $\beta_i$ and is calculated as following:

$$\beta_i(C) = \frac{\alpha * C_i^{SIM,CY}}{\alpha * C_i^{SIM,CY} + (1-\alpha) * C_i^{SIM,MT}} \qquad (12)$$

Considering that the simulated net flux through the $i$th reaction represents the difference between the forward and backward fluxes:

$$\upsilon_i^{net} = \upsilon_i^f - \upsilon_i^b \qquad (13)$$

and that the Gibbs free energy determines the forward-to-backward flux ratio (in accordance with the flux-force relationship; Eq. (14)):

$$r_i = \frac{\upsilon_i^f}{\upsilon_i^b} = Q^{-1} e^{-\frac{\Delta G^{\prime 0}}{RT}}, \qquad (14)$$

the forward and backward fluxes are calculated as following:

$$v_i^f = \frac{v_i^{net*} r_i}{r_i - 1} \quad (15)$$

$$v_i^b = \frac{v_i^{net}}{r_i - 1}$$

Given the estimated forward–backward fluxes, the unique MID of all metabolites in mitochondria and cytosol are computed via the Elementary Metabolite Unit (EMU)[39].

Direct solving of the above optimization problems is challenging due to the non-linear constraint in Eq. (11e), enforcing net flux in the direction of negative Gibbs free energy change. We describe a 4-step approach for solving this problem:

Step 1—Determine bounds on the direction of new flux based on thermodynamic analysis: We utilize mixed-integer linear programming (MILP) to search for net fluxes in the forward or backward direction (denoted $v^{net,f}$ and $v^{net,b}$, respectively), log metabolite concentrations (denoted $C$), and metabolite standard Gibbs free energies ($\Delta G^0$), satisfying mass-balance constraints and the second law of thermodynamics (as well physiological measurements)[40]. We define two variables for each reaction $i$, denoting whether net flux is in the forward ($y_i^f$) or backward ($y_i^b$) direction, enabling formulation of the thermodynamic constraint via linear equations:

$$\underset{y^f, y^b, v^{net,f}, v^{net,b}, C, \Delta G'^0}{\min/\max} \left( v^{net,f} - v^{net,b} \right) OR(C) OR \left( \Delta G'^0 + RT\ln Q \right) \quad (16)$$

$$s.t.$$

$$S v^{net,f} - S v^{net,b} = 0 \quad (16a)$$

$$v^{net,lb} \leq v^{net,f} - v^{net,b} \leq v^{net,ub} \quad (16b)$$

$$\ln[C]_{lb} \leq \ln[C] \leq \ln[C]_{ub} \quad (16c)$$

$y_i^f, y_i^b \in \{0,1\}$ (16d) for reactions i with thermodynamic data

$$\left( y_i^b - 1 \right) * B \leq \Delta G_i'^0 + RT\ln Q(C) \leq \left( 1 - y_i^f \right) * B \quad (16e)$$

$$v_i^{net,f} \leq T * y_i^f \quad (16f)$$

$$v_i^{net,b} \leq T * (1 - y_i^b) \quad (16g)$$

where Eq. (16e) constrains the sign of Gibbs free energy for the $i$th reaction based on $y_i$ and Eq. (16f–g) constrain flux in the forward and backward direction for the $i$th reaction based on $y_i$. The optimization problem is repeatedly applied to compute the range of feasible net fluxes, metabolic concentrations, and reaction Gibbs energies satisfying the above constraints (as in Flux Variability Analysis[79]).

Step 2—Determine bounds on the direction of new flux based on thermodynamic analysis: We formulate a variant of the optimization problem in Eq (11), in which the second law of thermodynamics (Eq. (11e)) is enforced only for reactions whose direction of net flux was inferred from the analysis performed in Step 1, using a linear constraint. The resulting nonconvex optimization with linear constraints is solved using sequential dynamic programming (SQP) starting from

1000 sets of random fluxes and metabolite concentrations to overcome potential local minima To facilitate efficient solving, we calculate the first order derivatives of the objective function ($ObjF$) with respect to each of the optimization parameters: The first order derivatives of the objective function with respect to forward ($\frac{dObjF}{dv_i^f}$) and backward ($\frac{dObjF}{dv_i^b}$) fluxes are computed as previously described in the EMU method[39]. These are used here to compute the derivative of the objective function with respect to the optimization problem variables: net fluxes ($v^{net}$), metabolite concentrations ($C$) and standard Gibbs free energies ($\Delta G^0$):

$$\frac{dObjF}{dv_i^{net}} = \frac{dObjF}{dv_i^f} * \frac{dv_i^f}{dv_i^{net}} + \frac{dObjF}{dv_i^b} * \frac{dv_i^b}{dv_i^{net}} = \frac{dObjF}{dv_i^f} * \frac{r_i}{r_i - 1} + \frac{dObjF}{dv_i^b} * \frac{1}{r_i - 1} \quad (17)$$

$$\frac{dObjF}{d(\ln C_j)} = \sum_i \left( \frac{dObjF}{dv_i^f} * \frac{dv_i^f}{d(\ln C_j)} + \frac{dObjF}{dv_i^b} * \frac{dv_i^b}{d(\ln C_j)} \right) = \sum_i sign\left( S_{ji} \right) \frac{v_i^{net*} r_i}{(r_i - 1)^2} \quad (18)$$

$$\frac{dObjF}{d(\Delta G_i'^0)} = \frac{dObjF}{dv_i^f} * \frac{dv_i^f}{d\left( \Delta G_i'^0 \right)} + \frac{dObjF}{dv_i^b} * \frac{dv_i^b}{d\left( \Delta G_i'^0 \right)} = \frac{2}{RT} * \frac{v_i^{net*} r_i}{(r_i - 1)^2} \quad (19)$$

We repeatedly applied the nonconvex optimization to compute the 95% confidence intervals for all fluxes and metabolite concentrations. The nonconvex optimization problem was solved using MATLAB's Sequential Quadratic Optimization (SQP), starting from 1000 sets of random fluxes and metabolite concentrations to overcome potential local minima. To compute confidence intervals for each compartmentalized flux, we started at the optimal solution, and increased (and then decreased) the value of the flux whose sensitivity is examined step-by-step. All other fluxes are determined by minimizing the objective function via the SQP, till the objective function has increased by 95% quantile of chi-square distribution with one degree of freedom, compared to the value of the optimal solution[80]. A similar method was used to compute the confidence intervals for each metabolite concentration in mitochondria and cytosol.

The inferred flux and concentration confidence intervals are used as input to iteratively run Step I and II, aiming to infer the direction of net flux through as many reactions as possible.

Step 3—Enumerate remaining thermodynamically feasible flux directionality vectors: In case the iterative method converged before all reaction directions are inferred, we enumerated all thermodynamically possible directionality vectors for the remaining reactions. This was done by repeatedly solving the MILP problem described in Step I (until no additional feasible solutions are found), while adding "integer cut" constraints to exclude previous solutions with the same set of net flux directions (while considering the derived ranges of net fluxes, concentrations, and Gibbs free energies derived in the previous iterative procedure). Specifically, in each $k$th run of the MILP problem, the following constraint was added:

$$\sum_j \left( \left| y^{f,j} - y_i^{f,j} \right| \right) \geq 1 \quad for \; i = 1,2,\ldots k - 1 \quad (20)$$

where $y_i^{f,j}$ denotes the integer variables denoting whether reactions are thermodynamically feasible in the forward direction in the ith run of the MILP problem.

Step 4—Compute flux and concentration confidence intervals considering all reaction directionality vectors:

For each feasible reaction directionality vector inferred in Steps 3, we solve the nonconvex optimization problem described in Step 2 (with the 2nd law of thermodynamics and flux-force relationship constraints now enforced for all relevant reactions in the model).

Finally, we determine confidence intervals for compartmentalized fluxes and metabolite concentrations based on the union of the ranges computed for all directionality vectors. The derived confidence intervals represent the level of uncertainty in flux estimates, while some intermediate flux values within those ranges might be infeasible.

Previous MFA studies have evaluated the goodness-of-fit between the measured and simulated metabolite isotopic labeling forms, based on the variance-weighted sum of squared residuals[39]. Here, we extended this analysis, to also similarly fit the simulated and measured metabolite concentrations (see Eq. (11)). The sum of square residuals of the labeling forms and concentrations is a stochastic variable with chi-square distribution. The number of degrees of freedom equal to the number of independent measurements minus the number of estimated free fluxes and metabolite concentrations. We determined statistical significance based on 95% quantile of chi-square distribution, obtaining a statistically significant fit with the experimental measurements in the application of CODE-MFA on all studied cell lines (chi-square $P$ value < 0.05).

### Reporting summary

Further information on research design is available in the Nature Portfolio Reporting Summary linked to this article.

## Data availability

The authors declare that all the data supporting the findings of this study are available within the article and its Supplementary Information and from the corresponding author upon request. Source data are provided with this paper.

## Code availability

All code used to conduct the research detailed in this manuscript is available at the following link: https://github.com/sternal75/CODE-MFA.

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

## Acknowledgements
The research leading to these results has received funding from the European Research Council (ERC) grant agreement no. 714738.

## Author contributions
A.S. and T.S. conceived and designed the study. M.F., B.S., A.A.A., W.D.L., N.S., and E.A. performed experiments. M.F., B.S., A.A.A., W.D.L., and E.A. performed LC-MS and stable isotope tracing measurements. M.F. performed shRNA-mediated gene knockdown. A.S. and T.S. analyzed the results. A.S. and T.S. wrote the manuscript. This project was supervised by T.S. All authors participated in editing and refining the analysis and the final version of this manuscript.

## Competing interests
The authors declare no competing interests.
