## [Peer Review File · Nature Communications]

Inferring mitochondrial and cytosolic metabolism by coupling isotope tracing and deconvolutionReviewer #1 (Remarks to the Author):

The study of Stern et al. introduces CODE-MFA, a computational approach to estimate compartmentalized reaction fluxes and metabolite concentrations in cytosol and mitochondria by integration and deconvolution of labelling patterns from isotope tracing experiments. The approach combines thermodynamic metabolic flux analysis (TMFA), that allows fixing the direction of reactions based on data on standard Gibbs free energy of reactions and concentrations of metabolites with, now, standard techniques from metabolic flux analysis (MFA), relying on simulation of labelling patterns based on elementary metabolite units (EMUs). The claim is that this approach provides more precise estimates – a characteristic that is particularly useful in the comparison of fluxes across different scenarios (in this study, different cell lines).

The reviewer focused on assessing the modelling and statistical aspects of the work; few comments are also provided about the biological implications of the findings. The work is of interest to the MFA and constraint-based modelling community. However, (i) clarifications and precise statements of the assumptions need to be provided to critically assess the implications of the comparisons made; (ii) statistical assessments of the fits are entirely missing, and must be provided for a fair and complete assessment of the considered scenarios, and (iii) differences in the labelling patterns in compartmentalized metabolite pools need to be shown and investigated. Finally, and most importantly, the CODE-MFA fits of the compartmentalized metabolite pools should be compared against independent measurements from non-aqueous fractionation or other independent technique, to further demonstrate the added value of the approach.

In the following, the reviewer provides the points of concern, divided into major and minor.

Major comments

Methods – inferring mitochondrial and cytosolic redox co-factor ratios

- $\alpha^{\{CY\}}$ and $\alpha^{\{MT\}}$ are mentioned in the text after Eqs. (6a) – (6d), but they do not appear in these equations; shouldn't this be just α for relative volume of the cytosol?

- The sentence "To compute the range of feasible mitochondrial NADP⁺/NADPH ratio, ..." does not make sense.

- Beta's are not defined in Eqs. (8) and (9). They are defined later in Eq. (12), so some restructuring is needed. In addition, gamma appears in several expressions in the methods, but it is nowhere defined.

- It is not clear if in this analysis errors for the standard Gibbs free energy of MPC is considered? If not, this should be included and the new results shown and critically assessed. The same holds for the findings based on Eq. (10).

Methods – CODE-MFA

- The forward and backward fluxes are obtained from Eqs. (14) – (16), which require that Q is known or can be estimated. Since Q can vary for a fixed net flux, the derivation of the ranges for the forward and backward fluxes should be carefully detailed and explained.

- What does "SQP was solved iteratively whole constraining the flux to increasing (and then decreasing) values" mean? SQP has a precise formulation in terms of updating the initial guess by using second-order Taylor approximation (see Wiechert's early work). This requires further detailing.

- In Step 4, what happens in the determination of the confidence intervals, by the suggested strategy in the case that the union of the ranges is disconnected? The reviewer is not convinced of the appropriateness of this approach.

Introduction

- Works on non-aqueous fractionation and their usage in flux estimation in eukaryotic cells should be mentioned.

- The reviewer finds that the point about assumptions of same labelling patterns for a metabolite in different compartments needs to be further stressed. This should motivate additional analyses, to show if and to what extent the simulated labelling patterns in the compartmentalized pools based on the inferred flux distributions differ.

Results

- R5P stands for ribulose-5-phosphate; is ribulose-5-phosphate that appears in several

places of the text and figures a typo?

- Reference to methods is given to explain that the ratio of forward and backward fluxes of 6PGD are given by $RP5_{m+5} / 6PG_{m+5}$, but details are not provided in the methods. The main assumption is that R5P is transformed only by 6PGD in vivo! How valid is this assumption? A critical discussion on this issue should be provided.

- To estimate the ratio of NADP+ and NADPH in the cytosol, by Eq. (3), intracellular CO2 concentration must be available; however, this is not measured (per Suppl. Table 7), so how exactly was this ratio determined (outside of CODE-MFA)? No details are provided in the methods, and the reviewer suspects that the uncertainty in the estimates of the ratio and the compartmentalized pools will become considerably larger than shown in Fig. 1.

- The current estimates on Fig. 1 seem to neglect the uncertainty in the Gibbs free energy of the considered reactions in deriving the compartmentalized concentrations and ratios, but these do not seem to be considered. What would factoring in this effect imply, with respect to comparison with the existing estimates from other methods, mentioned by the end of the section?

- Fig. 2d is a schematic, not obtained from an actual run; thus, it should not be used to support the stamen where it is first referenced.

- Measures for the goodness-of-fit are missing, and must be included before providing in-depth analysis of the 95% intervals for the net fluxes. See my comments to the methods above, about deriving 95% intervals for the forward and backward fluxes. This will support the statements about "good match". Such statistical assessment is also not provided for CODE (without MFA).

- The comparison to MFA is unfair, since MFA does not fit an objective function. Perhaps fitting the measured concentrations in conjunction with the MFA constraints would result in fixing the direction of more reactions?

- Provide an assessment for the difference in the labelling patterns simulated based on the best fitting flux distribution, to show that there are differences in the labelling patterns for particular classes of metabolites.

- There is no explanation as to why the correlation between fluxes and transcripts is higher than the correlation between fluxes and enzyme abundances. Is this not rather problematic, given that enzyme abundances are "closer" to the fluxes?

- Most importantly, for the fitted subcellular concentrations, independent validation should be provided. This is needed since it is well known that the fits strongly depend on the used model (along with the ranges for the standard Gibbs free energy of the modelled reactions). Additional support will strengthen the outcomes from the fitting exercises.

- The same concerns about "good matches" appear in the section with the flux estimates for the three cell lines; statistical assessment needs to be provided.

Figures

- Why are there no confidence intervals for the simulated concentrations in Fig. 3b,c.

Minor comments

Methods – inferring mitochondrial and cytosolic redox co-factor ratios

- Fix "... was used compute ... "

- Fix "... the direction of net flux through as -- much -- reactions as possible."

Results

- Fix "... is within less than one standard deviation -- off -- the ... "

Reviewer #2 (Remarks to the Author):

In this paper Stern et al., use isotope labeling experiments and modeling in thermodynamics and metabolism in an attempt to assert specific differences in mitochondrial and cytosolic metabolism. Parsing the complexity of metabolic flux in higher cells is difficult given compartmentalized biochemistry into organelles. Overall, the authors cite volumetric measurements of the whole cell and mitochondria in their

algorithm to “deconvolute” mitochondria-specific enzyme metabolism. They compare this to FBA calculations with and without thermodynamics.

Resolving mitochondrial and cytosolic metabolism is an extremely arduous task to gather experimental information and perform a complex calculation. This paper comes over as a merely technical exercise to demonstrate that it is possible. However, it seems that numerous shortcuts were taken. The analysis rests on a massive number of assumptions which are either not justified or maybe just not properly cited. To be credible, the paper should describe all assumptions and critically evaluate their impact on the outcome.

1. The mitochondrial matrix volume is fundamental. The authors cite a paper by WW Chen et al, 2017 incorrectly. Chen et al. put mitochondrial volume at 6% of the cell volume, and not 20% as the authors say here in this manuscript. Furthermore, Chen et al note that approximately 60% of the mitochondrion is matrix by volume, i.e. around 3-4%. With 20%, Stern et al grossly overestimate the mitochondrial volume, and therefore mitochondrial concentrations of metabolites. This has a major impact on the entire analysis. Regardless of assuming a specific value for alpha, it is important to verify robustness of the results over a range of alphas. Unless corrected and validated, these results are not publishable.
2. The “simple approach to determine redox cofactor concentrations” using 6PG and R5P should be validated first, i.e. in uncompartimentalized cells. There are issues that are not discussed, like the presence of $^{13}\text{C}\text{O}_2$, or R5P with different labeling patterns as assumed.
3. The lactate feeding is likely to affect NAD^+/NADH concentrations and fluxes in lower glycolysis and TCA cycle. How can it be assumed that the NAD^+/NADH estimates hold for the non-lactate condition? Assumptions about cellular pyruvate and its compartmentalized concentrations need to be cited. There is no evidence provided from the authors to back up their assumptions.
4. CODE-MFA doesn't seem new at all. The implementation given by the Heinemann lab (Ref 48) for Step II seems more mature. How was CODE-MFA validated? This is critical because the software does not seem to be available. Details on convergence, multi-start, use of covariance matrix for measurements, etc etc are needed.
5. Apart of glucose and glutamine, the medium contains plenty of nutrients. How were they quantified?
6. The authors claim that this is a “first quantitative view of mitochondrial and cytosolic central energy fluxes”. This is wrong as previous works has done this using digitonin solubilized cells.
7. It is unclear why the authors present FPKM data in the main text figure, but then use protein-level information in the supplemental figure. Why isn't the protein-level expression data shown as a main figure as enzymes and the supposed biology of this study are operative at the protein-level?
8. The authors inappropriately use “infer” throughout the manuscript. To infer is to deduce from reasoning rather than explicit experiments. However, the authors describe their labeling experiments as inferences, and use direct biochemical evidence from labeling patterns for this manuscript. The authors should be more careful with their language in references to the experiments that were performed. For example, in their discussion they say this is the “first time to directly quantify mitochondrial and cytosolic fluxes and metabolite concentrations”, but they infer these results and do not directly quantify fluxes....
9. It is unclear why the NADP/NAD section is included in this manuscript. How relevant is the measurements for the determination of fluxes? An ablation experiment is needed.
10. Many figures misrepresent biology (mislabeling of mitochondrial transporters and their mechanisms in figure 5), or do not provide real insight or support the claims of this paper. For example, Figures 2 and 3 could be supplemental information and support some biological insight presented as a main figure.
11. The authors do not comment at all on why specific cancer cell lines were chosen, particularly as one is KRAS mutant, and expected to have a different metabolic phenotype. The conclusions drawn from cell line comparisons are not insightful and not

validated. The authors note that for some cases that data is consistent with previously published, measured data. However, consistency with published findings does not represent any real novelty.

12. The LC-MS methods section presented by the authors are highly problematic. For example they say "Concentrations of metabolites were determined with the standard addition and isotope ratio methods using chemical standards". This data/information is not presented nor cited. Unclear why they provide upper and lower bounds of metabolite concentrations and not the mean and standard deviations of their measurements. Also unclear how absolute concentrations were calculated and if SIL data is actually in a linear and quantifiable range. The methods make no reference to a specific way to calculate this and it is unclear why and which data was collected by several different LC methods. Was a SIM scan the only method used for MS data collection? Or only for specific metabolites? Was a standard curve generated for each SIM scan to calculate absolute concentrations, because absolute concentration ranges are provided? There is no information about AGC values used for analytical measurements and the width of SIM scans is not provided. There is also no specific information about data analysis nor how LC-MS peaks and/or isotopologues were quantified in this study outside of the software that was used. It's difficult for this reviewer to believe that they can accurately detect and quantify lactate isotopologues and the host of other low abundance isotopologues given the methods provided in the manuscript given the methods provided.

13. The proof that isotope tracing studies were at steady state must be provided in the supplement.

14. S11 seems to use the wrong mitochondrial pH to calculate Gibbs free energy. There is no reference for why pH 7.5 is used. Most thermodynamic modeling papers have used pH 8.5. This reflects primary biochemical evidence from a number of studies.

15. The authors note in the introduction that a number of papers have used direct measurements of mitochondrial metabolites or have analytically determined the NAD⁺/NADH ratio in cells. However, in the manuscript there is no comparison back to these cited papers about how their algorithm compares. This would be valuable information if the authors want to draw any conclusive, comparative results about their method, and their claims that this algorithm will actually be of utility for the field. The authors do not provide sufficient evidence outside of comparisons to other FBA algorithms to demonstrate method utility or application to published data and exemplary cases. There does not seem to be any effort to relate to information from the field.

Minor points:

- Figure legends are chronically mislabeled and inaccurate.
- For figure 1, the reaction diagrams are incorrectly drawn with a one-sided arrow if the authors would like to claim reaction reversibility.
- Figure 2 is not insightful.
- The number of significant figures provided in the supplemental tables is incorrect relating values and standard deviations.
- The authors note in their introduction that "...we show how mitochondrial and cytosolic metabolic activities can be inferred based on metabolic measurements performed on intact cells under physiological conditions by combining metabolic modelling and computational deconvolution."
- This claim implies that their method accurately reflects the underlying biology, but they do not test/validate this claim.
- Figure 4 "measured" is actually calculated/inferred
- Why wasn't glucose labeling shown for figure 5? Only glutamine labeling is shown?

Reviewer #3 (Remarks to the Author):

This paper describes a computational approach to deconvolute stable isotope metabolic

flux data with the aim to visualise and understand metabolic compartmentalisation between the cytosol and mitochondria. This concept is interesting and novel, and I believe it would be of interest to readers of this journal. I have a few specific comments:

My main point is how to validate that the results from the model are correct. Currently the model looks to work well, but there appears to be no hard evidence validating whether it works or not. Could you run the model on multiple different phenotypes and then observe the changes to see how they differ to wild type? This could be picking 3 or so well defined changes that are biochemically well characterised in the literature that have mitochondrial and cytosolic components (e.g. hypoxic response, specific metabolic mutations, chemical metabolic inhibitors). Set some biological data in the lab, then run your model showing the output in a similar format as Figure 5. Then look to see if the biochemical changes you predict are the same as those previously reported in the literature. This would go a long way to showing that the method can accurately predict compartmental metabolic changes.

Further to the point above, it would be ideal to show your methodology on less well characterised (and thus novel) and medically relevant changes to see how these alter phenotype. In the discussion you allude to the value of your approach to SHMT1 or IDH mutations. Why not get a biological model with these mutations in the lab and apply your approach to observe and report novel metabolic changes.

The text in paper is complex (particularly the results section), and while I appreciate that it is a complex concept, could you expand some of the results to make it more accessible to the type of scientist who may wish to use this work? i.e. I believe many potential users of this approach will not be experts in mathematical modelling, so could these concepts be further explained in a way that is easier to understand?

When describing the results in the Hela cells, the pathway figure shown in Figure 5 is useful to the reader. However, this appears to only show one phenotype. What would be really useful is to show multiple phenotypes (see my point above). For example perturb the pathway in a known way (by using a metabolic inhibitor or gene alteration) and then observe the results to see how different they are from the resting phenotype. This will help to show that the model works to describe expected biochemical changes.

It's not totally clear how the first section of the results ("Inferring mitochondrial and cytosolic redox co-factors...") fits with the later sections of the results. Are the NAD/NADH and NADP/NADPH ratios used as an example of how such computational compartmentalisation can be done? Or do these specific reactions form a more fundamental importance for the whole CODE-MFA platform?

Are you able to show if metabolic reactions in other cellular compartments, e.g. peroxisomes, influence the cytosolic/mitochondrial balance? If these are unaccounted for, could they cause errors in the prediction of cytosolic/mitochondrial balance?

Considering each metabolic reaction you investigate in the results section, is there is an assumption that this reaction accounts for all the observed change in the metabolites from this reaction? For example in the case of NADPH, the levels appear to be inferred from changes to 6PG & R5P, thus specifically changes to the pentose phosphate pathway. How do you therefore account for changes to other processes that either contribute to NADPH/NADP levels, e.g. folate metabolism, fatty acid metabolism etc. If these are missed out from the model could the results you show be incorrect? This comment should also be addressed for the other reactions shown in the paper.

Reviewer #4 (Remarks to the Author):

The idea of directly quantify mitochondrial and cytosolic fluxes and metabolite

concentrations via measurements performed with intact cells under physiological conditions (without requiring subcellular fractionation) is new and challenging. However the thermodynamic assumptions and the over simplification of the model in terms of the reactions that are considered as well as on the thermodynamic constraints rise important concerns on the model predictions.

In the following the major concerns identified:

1) It is not properly justified that the cytosolic NADP/NADPH balance can be inferred directly from 6PGD activity taking into account that exist other important contributors to the cytosolic NADP/NADPH balance such as NAD⁺ kinase (NADK) (a cytosolic enzyme that can generate NADP and that it has been demonstrated that inhibition of NADK impact NADPH levels Tedeschi PM, Lin H, Gounder M, et al. *Suppression of Cytosolic NADPH Pool by Thionicotinamide Increases Oxidative Stress and Synergizes with Chemotherapy*. *Mol Pharmacol*. 2015;88(4):720-727. doi:10.1124/mol.114.096727). ME1 and MTHFD1 are also NADP/NADPH dependent enzymes in the cytosol.

2) The authors assume that forward-backward flux through 6PGD can be inferred from M+5 6PG assuming that it is synthesized only through reverse 6PGD flux from R5P (incorporating 12CO₂). Taking into account that 6PGD affinity by CO₂ is very low and that M+5 6PG can be generated from different carbon rearrangements including central metabolism, reversible non-oxidative PP etc... This assumption need to be validated experimentally.

3) The central metabolism model not include important reactions that exchange C₆, C₆ and C₃ pools which can have a high impact on mass isotopomers distribution. For instance, the authors not include in the model the non-oxidative branch of PPP which connects pools of C₆ with C₅, C₃..... This pathway need to be included as it is completely reversible and has a high impact on ¹³C label distribution among central metabolism metabolites.

4) In the excel file input.xlsx file (reaction list part of the model inputs), it is written that V2 can only go on the direction Glutamate_media_CY => OTHER_5_carbon Being the lower and upper limits 0,1 to 70 . This means that it is assumed that only a flux of entry of glutamate inside the cell is permitted in the model. This is unrealistic taking into account that in many cell models glutamate is mainly released from the cell to the extracellular media.

In the .pdf supplementary material file (Table 1) the reaction v2 is written in the reverse direction when compared with the same reaction v2 in the input.xlsx file . The upper and lower limits are both positive which means that according Table 1 v2 can only in the direction cell -> media and in the input.xlsx the same v2 reaction can only go in the reverse direction media->cell which is contradictory.

5) In the excel file input.xlsx (metabolites part of the model input) appear metabolites such as "6phosphogluconate_MT" that not appear in the reaction list. In addition 6-phosphogluconate is only cytosolic and not need to be defined as _MT. There are other examples of metabolites in the input excel (metabolites sheet) that not appear as entities involved in the reaction list.

6) The authors infer the cytosolic NAD⁺/NADH ratio by analyzing the forward-backward flux of cytosolic lactate dehydrogenase (LDH), feeding cells with [U-¹³C]-lactate, and measuring the total cellular concentration of pyruvate and lactate (Figure 1e-h). Taking into account that NAD⁺/NADH depends on the available substrates and that are other reactions that can contribute to this ratio, this inference need to be further justified or validated.

7) Additional evidence is necessary to validate the method. Comparison of the proposed

method with other methods in the literature using cell fractionation need to be done. Also it will be convenient to compare results with control cells and cells in which key players in the model are knockout to see if the model is able to do an accurate prediction of the decreased flux through the corresponding reaction.

In the present form the work is not acceptable for publication as additional evidence is necessary to support the conclusions. The assumptions of the model and the reactions considered also need to be justified as important reactions in carbon metabolism that impact in label Distribution are not included in the model. A major revision is required. To reproduce the work it is necessary to have matlab. I do not have matlab in my lab so I cannot try to reproduce. The scripts and files provided seem complete but there are some inconsistencies between Table 1 (supplementary data) and input.xlsx on the directionality of the reactions and the upper and lower bounds used that raise some concerns as detailed in the previous comments. Also it seems inconsistent that in the metabolites sheet (excel input.xlsx file) appear metabolites that are not used in any reaction.

Reviewer #1:

The study of Stern et al. introduces CODE-MFA, a computational approach to estimate compartmentalized reaction fluxes and metabolite concentrations in cytosol and mitochondria by integration and deconvolution of labelling patterns from isotope tracing experiments. The approach combines thermodynamic metabolic flux analysis (TMFA), that allows fixing the direction of reactions based on data on standard Gibbs free energy of reactions and concentrations of metabolites with, now, standard techniques from metabolic flux analysis (MFA), relying on simulation of labelling patterns based on elementary metabolite units (EMUs). The claim is that this approach provides more precise estimates – a characteristic that is particularly useful in the comparison of fluxes across different scenarios (in this study, different cell lines).

The reviewer focused on assessing the modelling and statistical aspects of the work; few comments are also provided about the biological implications of the findings.

1. *“The work is of interest to the MFA and constraint-based modelling community. However, (i) clarifications and precise statements of the assumptions need to be provided to critically assess the implications of the comparisons made;”*

See “The underlying assumptions of CODE-MFA” on page 1.

2. *“(ii) statistical assessments of the fits are entirely missing, and must be provided for a fair and complete assessment of the considered scenarios”*

We assessed the goodness of fit based on the method described by Antoniewicz et al²². Briefly, our optimization method minimizes the variance-weighted sum of squared residuals between the measured and simulated metabolite isotopic labeling forms and concentrations (Eq. 11); a stochastic variable with a chi-square distribution. Statistical significance was determined based on 95% quantile of chi-square distribution, considering the number of degrees of freedom (see Methods). We now describe this statistical analysis in more detail, showing that the application of CODE-MFA to all studied cell lines provided a statistically significant fit with the experimental measurements (chi-square p-value < 0.05). We further performed rigorous statistical analysis of confidence intervals for inferred compartmentalized fluxes, concentrations, and Gibbs energy, as described in Methods.

3. *“(iii) differences in the labelling patterns in compartmentalized metabolite pools need to be shown and investigated.”*

Our analysis shows that for as much as 20% of the metabolites in the model, there is a major difference (>20%) in the relative abundance of a mitochondrial and cytosolic isotopic labeling form, while feeding HeLa cells with [U-¹³C]-glucose or [U-¹³C]-glutamine (Figure L3). We discuss several examples for such differences in compartmentalized labeling patterns (see Figure 4b-d). Feeding [U-¹³C]-glutamine, rapid forward-backward flux through PCK2 results in the synthesis of mitochondrial M+3 phosphoenolpyruvate (PEP; Figure 4c) and M+3 OAA (releasing ¹³CO₂ from M+4 OAA and fixing atmospheric ¹²CO₂): The

mitochondrial M+3 PEP is not observed in total cell measurements, due to the substantially larger PEP pool size in cytosol (Figures 4c and 3d); while M+3 OAA condenses with non-labelled acetyl-CoA via citrate synthase, producing the experimentally observed M+3 citrate (Figure 4b). The M+3 OAA further goes through the malate-aspartate shuttle, synthesizing mitochondrial M+3 aspartate, cytosolic M+3 OAA, and then cytosolic M+3 malate. This leads to higher fractional labeling of M+3 malate in cytosol (21%) than in mitochondria (10%; Figure 4d), while the convoluted labeling of M+3 malate (19%) is biased towards that in cytosol (where malate pool size is large), in agreement with the experimental measurement (Figure 3d).

Figure L3: Cumulative distribution of the maximum difference in metabolite isotopic labeling form in cytosol and mitochondria, while feeding HeLa cells with [U-¹³C]-glucose or [U-¹³C]-glutamine. X-axis represents the difference in isotopic labeling form abundance between the two compartments (maximum value across all isotopic labeling forms of a metabolite); Y-axis represents the fraction of metabolites in the model.

4. *“Finally, and most importantly, the CODE-MFA fits of the compartmentalized metabolite pools should be compared against independent measurements from non-aqueous fractionation or other independent technique, to further demonstrate the added value of the approach.”*

See “Validation of CODE-MFA compared to state-of-the-art methods and currently known compartmentalized metabolic quantities” on page 3.

In the following, the reviewer provides the points of concern, divided into major and minor.

Major comments

Methods – inferring mitochondrial and cytosolic redox co-factor ratios

5. *“ α^{CY} and α^{MT} are mentioned in the text after Eqs. (6a) – (6d), but they do not appear in these equations; shouldn’t this be just α for relative volume of the cytosol? “*

We thank the reviewer for noticing that. Text was changed accordingly.

6. *“The sentence “To compute the range of feasible mitochondrial NADP+/NADPH ratio, ...” does not make sense.”*

We rephrased the sentence: “To compute the range of **possible** mitochondrial NADP+/NADPH ratios, ...”

7. *“Beta’s are not defined in Eqs. (8) and (9). They are defined later in Eq. (12), so some restructuring is needed. In addition, gamma appears in several expressions in the methods, but it is nowhere defined.”*

We restructured the equations as following: We replaced beta with alpha in eq. 8 and 11. Alpha is denoted in eq. 4 and 5, as the relative volume of the cytosol out of the total cellular volume. We also removed eq. 12 and 13.

8. *“It is not clear if in this analysis errors for the standard Gibbs free energy of MPC is considered? If not, this should be included and the new results shown and critically assessed. The same holds for the findings based on Eq. (10).”*

We considered a standard deviation of 30% in the estimated Gibbs free energy of MPC transporter, as well as to all other transporters, calculated per equation 10 (see Table S11).

Methods – CODE-MFA

9. *“- The forward and backward fluxes are obtained from Eqs. (14) – (16), which require that Q is known or can be estimated. Since Q can vary for a fixed net flux, the derivation of the ranges for the forward and backward fluxes should be carefully detailed and explained. “*

Equation 11 describes the CODE-MFA optimization problem formulation. We search for the most likely mitochondrial and cytosolic net fluxes under metabolic steady-state, compartmentalized metabolite concentrations, and reactions’ standard Gibbs free energies, providing optimal match with concentration and isotopic labelling measurements performed in intact cells. Note, that the optimization parameters include net fluxes, metabolite concentrations (determining Q), and standard Gibbs free energies. With these three optimization parameters, we calculate the forward and backward fluxes, as described in equations 14-16 (changed to 12-14 in the revised manuscript); used to compute the unique MIDs of all metabolites via the Elementary Metabolite Unit (EMU).

10. *“What does “SQP was solved iteratively while constraining the flux to increasing (and then decreasing) values” mean? SQP has a precise formulation in terms of updating the initial guess by using second-order Taylor approximation (see Wiechert’s early work). This requires further detailing.”*

Our intention was not to explain how SQP works, but to explain that the confidence intervals for each compartmentalized flux is determined by iteratively constraining the flux to increasing and then decreasing values. For each constrained flux value, the non-convex optimization is solved via a standard

SQP solver (using the MATLAB function `fmincon` from the MATLAB optimization toolbox). To make it clear, we rephrased the above to:

“To compute confidence intervals for each compartmentalized flux, we started at the optimal solution, and increased (and then decreased) the value of the flux whose sensitivity is examined step-by-step. All other fluxes are determined by minimizing the objective function via the SQP, till the objective function has increased by 95% quantile of chi-square distribution with one degree of freedom, compared to the value of the optimal solution.”

11. *“In Step 4, what happens in the determination of the confidence intervals, by the suggested strategy in the case that the union of the ranges is disconnected? The reviewer is not convinced of the appropriateness of this approach.”*

The derived confidence intervals represent the level of uncertainty in flux estimates, while some intermediate flux values within those ranges might be infeasible (this is now specified in the Methods). Still, the unified confidence intervals derived by CODE-MDA are smaller than those of MFA by an order-of-magnitude.

Introduction

12. *“Works on non-aqueous fractionation and their usage in flux estimation in eukaryotic cells should be mentioned.”*

We added a reference for non-aqueous fractionation (NAF) in the manuscript. Though, notice that this technique is widely used in plant science, and no appropriate separation of the mitochondrial compartment has been achieved using this method^{14,15}.

13. *“The reviewer finds that the point about assumptions of same labelling patterns for a metabolite in different compartments needs to be further stressed. This should motivate additional analyses, to show if and to what extent the simulated labelling patterns in the compartmentalized pools based on the inferred flux distributions differ.”*

This was now further assessed. See above reply (and Figure L3)

Results

14. *“R5P stands for ribulose-5-phosphate; is ribulose-5-phosphate that appears in several places of the text and figures a typo? “*

Typo fixed.

15. *“Reference to methods is given to explain that the ratio of forward and backward fluxes of 6PGD are given by $R5P_{m+5} / 6PG_{m+5}$, but details are not provided in the methods. The main assumption is that R5P is transformed only by 6PGD in vivo! How valid is this assumption? A critical discussion on this issue should be provided.”*

We did not assume that R5P is metabolized only through 6PGD, as R5P can indeed be synthesized through the non-oxidative PPP. We quantified the forward-backward fluxes through 6PGD by measuring the m+5 labeling of R5P ($R5P_{m+5}$) and 6PG ($6PG_{m+5}$) when feeding HeLa cells with [U-¹³C]-glucose, using SeQuant ZIC-pHILIC column (0.98 and 0.05 respectively; Table S2), where isotopic balance entails that the ratio is equal to $\frac{R5P_{m+5}}{6PG_{m+5}}$. The fact that $R5P_{m+5}$ is smaller than 1 indicates that a reaction other than 6PGD also contributes to synthesizing R5P.

16. *“To estimate the ratio of NADP+ and NADPH in the cytosol, by Eq. (3), intracellular CO₂ concentration must be available; however, this is not measured (per Suppl. Table 7), so how exactly was this ratio determined (outside of CODE-MFA)? No details are provided in the methods, and the reviewer suspects that the uncertainty in the estimates of the ratio and the compartmentalized pools will become considerably larger than shown in Fig. 1. “*

We did not measure the concentration of CO₂, but used a common assumption of 1.2mM^{23,24}. We repeated the CODE-MFA analysis of HeLa cells to confirm the robustness of our results to 3-fold lower and 3-fold higher CO₂ concentrations (0.4mM and 3.6mM). Indeed, we found that the inferred compartmentalized fluxes were highly similar to those inferred using the original 1.2mM concentration, with a pairwise Pearson correlation > 0.98 (Pearson *p*-value < 10⁻¹⁰).

17. *“The current estimates on Fig. 1 seem to neglect the uncertainty in the Gibbs free energy of the considered reactions in deriving the compartmentalized concentrations and ratios, but these do not seem to be considered. What would factoring in this effect imply, with respect to comparison with the existing estimates from other methods, mentioned by the end of the section?”*

Our estimations of the standard Gibbs free energies do not neglect the Gibbs free energy uncertainty. The standard Gibbs free energies, including the uncertainties are estimated by the group contribution method²⁵, directly obtained from the equilibrator website (<http://equilibrator.weizmann.ac.il>).

18. *“Fig. 2d is a schematic, not obtained from an actual run; thus, it should not be used to support the statement where it is first referenced.”*

Reference changed to Figure S1b, describing the number of reactions whose direction of net flux is uniquely inferred across CODE-MFA iterations in HeLa cells.

19. *“Measures for the goodness-of-fit are missing, and must be included before providing in-depth analysis of the 95% intervals for the net fluxes. See my comments to the methods above, about deriving 95%”*

intervals for the forward and backward fluxes. This will support the statements about “good match”. Such statistical assessment is also not provided for CODE (without MFA).”

See response 2 above.

20. *“The comparison to MFA is unfair, since MFA does not fit an objective function. Perhaps fitting the measured concentrations in conjunction with the MFA constraints would result in fixing the direction of more reactions?”*

Indeed, MFA fits only isotopic labeling patterns and not metabolite concentrations. We further tested an additional control in which we first aimed to determine the direction of net flux based on thermodynamic analysis (Step 1 in CODE-MFA) prior to running MFA. However, considering thermodynamics alone was insufficient for determine net flux directionality, thus could not improve the standard MFA results.

21. *“Provide an assessment for the difference in the labelling patterns simulated based on the best fitting flux distribution, to show that there are differences in the labelling patterns for particular classes of metabolites.”*

See above reply (and Figure L3).

22. *“There is no explanation as to why the correlation between fluxes and transcripts is higher than the correlation between fluxes and enzyme abundances. Is this not rather problematic, given that enzyme abundances are “closer” to the fluxes?”*

We compared the fluxes inferred by CODE-MFA to enzyme abundances from another paper (Itzhak et al.²⁶), and found higher correlation (Spearman correlation = 0.64; p-value = 0.01) than the correlation we presented in the original manuscript in figure S4c (Nagaraj et al.²⁷; Spearman correlation = 0.54; p-value = 0.03). Yet, the correlation between fluxes and transcripts is still higher (Spearman correlation = 0.72; p-value = $5 \cdot 10^{-4}$). The lower correlation between fluxes and enzyme levels can be explained by the noisy nature of these measurements – testified by the even lower correlation between the actual enzyme concentration measurements performed by the two studies (Spearman correlation = 0.57; p-value = 0.01).

23. *“Most importantly, for the fitted subcellular concentrations, independent validation should be provided. This is needed since it is well known that the fits strongly depend on the used model (along with the ranges for the standard Gibbs free energy of the modelled reactions). Additional support will strengthen the outcomes from the fitting exercises.”*

See “Validation of CODE-MFA compared to state-of-the-art methods and currently known compartmentalized metabolic quantities” on page 3, and “A new validation study for the importance of the inferred high cytosolic malic enzyme flux in one of the studied cell lines” on page 4; and response 4 above.

24. *“The same concerns about “good matches” appear in the section with the flux estimates for the three cell lines; statistical assessment needs to be provided.”*

See response 2 and 19 above.

Figures

25. *“Why are there no confidence intervals for the simulated concentrations in Fig. 3b,c.”*

We now changed figure 3b and 3c (changed to 2d and 2e in the revised manuscript) to include simulated metabolite concentrations and isotopic labeling with confidence intervals.

Minor comments

Methods – inferring mitochondrial and cytosolic redox co-factor ratios

26. *Fix “... was used compute ... ”*

Typos fixed.

27. *Fix “ ... the direction of net flux through as -- much -- reactions as possible.”*

Typos fixed.

Results

28. *Fix “... is within less than one standard deviation -- off -- the ... ”*

Typos fixed.

Reviewer #2:

In this paper Stern et al., use isotope labeling experiments and modeling in thermodynamics and metabolism in an attempt to assert specific differences in mitochondrial and cytosolic metabolism. Parsing the complexity of metabolic flux in higher cells is difficult given compartmentalized biochemistry into organelles. Overall, the authors cite volumetric measurements of the whole cell and mitochondria in their algorithm to “deconvolute” mitochondria-specific enzyme metabolism. They compare this to FBA calculations with and without thermodynamics.

1. *“Resolving mitochondrial and cytosolic metabolism is an extremely arduous task to gather experimental information and perform a complex calculation. This paper comes over as a merely technical exercise to demonstrate that it is possible. However, it seems that numerous shortcuts were taken. The analysis rests on a massive number of assumptions which are either not justified or maybe just not properly cited. To be credible, the paper should describe all assumptions and critically evaluate their impact on the outcome.”*

Indeed, inferring compartmentalized metabolism is extremely challenging. Our work presents, for the first time, an approach for directly quantifying mitochondrial and cytosolic fluxes and metabolite concentrations without perturbing the cells. Previous approaches demonstrated the potential of deconvoluting metabolite concentration measurements into compartmentalized pool sizes to infer compartmentalized metabolism^{28,29}, neglecting the bias due to potential difference in the isotopic labeling patterns in the mitochondria and the cytosol. Here we show how metabolite isotopic labeling patterns measured in isotope tracing experiments can be deconvoluted into distinct mitochondrial and cytosolic counterparts and facilitate direct inference of compartmentalized fluxes and metabolite concentrations.

We added a summary of all assumptions underlying CODE-MFA (See “The underlying assumptions of CODE-MFA” on page 1) and thoroughly evaluated the robustness of our analysis to various parameter choices (mitochondrial PH level, mitochondrial relative volume, and CO₂ concentration above; response 2 and 14 below, and response 16 to reviewer #1).

2. *“The mitochondrial matrix volume is fundamental. The authors cite a paper by WW Chen et al, 2017 incorrectly. Chen et al. put mitochondrial volume at 6% of the cell volume, and not 20% as the authors say here in this manuscript. Furthermore, Chen et al note that approximately 60% of the mitochondrion is matrix by volume, i.e. around 3-4%. With 20%, Stern et al grossly overestimate the mitochondrial volume, and therefore mitochondrial concentrations of metabolites. This has a major impact on the entire analysis. Regardless of assuming a specific value for alpha, it is important to verify robustness of the results over a range of alphas. Unless corrected and validated, these results are not publishable.”*

The relative cellular volume of mitochondrial matrix is estimated to be 4–20% across human cell lines (See “The underlying assumptions of CODE-MFA” - assumption 5 on page 3). We evaluated the robustness of our analysis performed for HeLa cells, considering a range of potential mitochondrial volumes (4%, 12%,

20%). Estimated compartmentalized fluxes were highly similar, with a pairwise Pearson correlation > 0.95 (Pearson p -value $< 10^{-10}$).

3. *“The “simple approach to determine redox cofactor concentrations” using 6PG and R5P should be validated first, i.e. in uncompartimentalized cells. There are issues that are not discussed, like the presence of $^{13}\text{CO}_2$, or R5P with different labeling patterns as assumed.”*

Analysis in uncompartimentalized cells is outside our scope. Our method makes no assumption regarding the isotopic labeling of 6PG and R5P, which are both measured; and no assumption is made regarding R5P being synthesized solely through the oxidative-PPP (See response 15 to reviewer #1). We do assume negligible amount of $^{13}\text{CO}_2$, which is supported by the non-detectable carbamoyl-aspartate m+1, when feeding isotopic glucose (Table S2).

4. *“The lactate feeding is likely to affect NAD^+/NADH concentrations and fluxes in lower glycolysis and TCA cycle. How can it be assumed that the NAD^+/NADH estimates hold for the non-lactate condition?”*

We assume that Lactate feeding does not affect NAD^+/NADH concentrations, as the added Lactate concentration (50uM) is significantly lower than that measured lactate concentration after 24h.

5. *“Assumptions about cellular pyruvate and its compartmentalized concentrations need to be cited. There is no evidence provided from the authors to back up their assumptions.”*

We make no assumptions of pyruvate concentrations; but rather measure the total cellular concentration and computationally derive its compartmentalized concentrations. Specifically, the compartmentalized concentration of pyruvate is based on thermodynamics analysis of the Mitochondrial Pyruvate Carrier (MPC), assuming net transport of Pyruvate from the cytosol to the mitochondria³⁰, dictating negative Gibbs free energy in this direction; and de-convolution of distinct mitochondrial and cytosolic concentrations to fit to the measured total cellular level (see equations 8 and 9 in the manuscript).

6. *“CODE-MFA doesn’t seem new at all. The implementation given by the Heinemann lab (Ref 48) for Step II seems more mature. How was CODE-MFA validated? This is critical because the software does not seem to be available. Details on convergence, multi-start, use of covariance matrix for measurements, etc etc are needed.”*

The Heinemann lab states in the paper that Ideally, one would simultaneously fit physiological, metabolome, and ^{13}C labelling data into a combined stoichiometric-thermodynamic-isotopomer model. However, they consider such integration highly challenging. The addition of the second law renders such model non-linear and the solution of mass and isotopomer models is already highly computationally demanding due to the large numbers of equations. Therefore, they used a 3-step approach: i) Fit a combined stoichiometric and thermodynamic model. ii) Sample net flux solutions from this solution space. iii) Fit the model to the ^{13}C -labeling data, while using a net flux sample as input, leaving the labelling

exchange fluxes as the sole free variables in the optimization. **Here, we describe an efficient algorithm to simultaneously fit physiological, metabolome, and ^{13}C labelling data into a combined stoichiometric-thermodynamic-isotopomer model.**

Furthermore, while the Heinemann paper utilize deconvolution of metabolite concentration measurements into compartmentalized counterparts, it does not account for potential differences in the isotopic labeling patterns in the mitochondria and the cytosol – which we show is highly abundant (See response 3 to reviewer #1). Recently, inference of compartment-specific isotopic labeling of metabolites, was achieved by isotope tracing followed by rapid cell fractionation, showing distinct isotopic labeling patterns in the mitochondria and the cytosol²⁴. Here we use metabolite isotopic labeling patterns measured in isotope tracing experiments and deconvolute the measured labeling information into distinct mitochondrial and cytosolic counterparts, practically not neglecting these distinct isotopic labeling patterns in the optimization problem.

Please refer to the comments regarding validation of our results (See “Validation of CODE-MFA compared to state-of-the-art methods and currently known compartmentalized metabolic quantities” on page 3, and “A new validation study for the importance of the inferred high cytosolic malic enzyme flux in one of the studied cell lines” on page 4) and regarding convergence and goodness of fit (See response 2 to reviewer #1). CODE-MFA code is now available at the following link:

<https://github.com/sternal75/CODE-MFA>

7. *“Apart of glucose and glutamine, the medium contains plenty of nutrients. How were they quantified?”*

Glucose and glutamine are known to be the predominant carbon sources in cultured cell lines. We determine no major carbon contribution from other nutrients based on intracellular metabolites labeling (See “The underlying assumptions of CODE-MFA” – assumption 3 on page 2).

8. *“The authors claim that this is a “first quantitative view of mitochondrial and cytosolic central energy fluxes”. This is wrong as previous works has done this using digitonin solubilized cells.”*

We changed the statement to express the fact that this is the first study which quantified compartmentalized fluxes and metabolite concentrations without cell fractionation – which could markedly bias metabolism due to non-physiological conditions; see example regarding previous bias in compartmentalized NADH measurement via cell fractionation.

9. *“It is unclear why the authors present FPKM data in the main text figure, but then use protein-level information in the supplemental figure. Why isn’t the protein-level expression data shown as a main figure as enzymes and the supposed biology of this study are operative at the protein-level?”*

See response 22 to reviewer #1.

“The authors inappropriately use “infer” throughout the manuscript. To infer is to deduce from reasoning rather than explicit experiments. However, the authors describe their labeling experiments as inferences, and use direct biochemical evidence from labeling patterns for this manuscript. The authors should be more careful with their language in references to the experiments that were performed. For example, in their discussion they say this is the “first time to directly quantify mitochondrial and cytosolic fluxes and metabolite concentrations”, but they infer these results and do not directly quantify fluxes....”

We used **infer** to highlight the fact that our method does not require a simplifying optimality assumption as in FBA; leading to what is typically referred to as flux **prediction**. We rephrased claims regarding “directly quantifying” and “directly probe” fluxes to say that we infer fluxes.

10. *“It is unclear why the NADP/NAD section is included in this manuscript. How relevant is the measurements for the determination of fluxes? An ablation experiment is needed.”*

The NADP/NAD section is included to provide more insight on how the method works instead of a mere ‘black-box’ description. It shows how the principles of flux/thermodynamics/de-convolution can be easily employed to gain important compartmentalized quantities with a minimal set of assumptions (i.e. without requiring a metabolic network model etc).

11. *“Many figures misrepresent biology (mislabeling of mitochondrial transporters and their mechanisms in figure 5), or do not provide real insight or support the claims of this paper. For example, Figures 2 and 3 could be supplemental information and support some biological insight presented as a main figure. “*

Figure 5 (changed to figure 4) provides a simplified view of central energy metabolism and obviously cannot show all information on enzyme/transporter names and participating co-factors. Figure 2-3 were combined with some sub-figures moved to supp. Material.

12. *“The authors do not comment at all on why specific cancer cell lines were chosen, particularly as one is KRAS mutant, and expected to have a different metabolic phenotype. The conclusions drawn from cell line comparisons are not insightful and not validated. The authors note that for some cases that data is consistent with previously published, measured data. However, consistency with published findings does not represent any real novelty.”*

See "Validation of CODE-MFA compared to state-of-the-art methods and currently known compartmentalized metabolic quantities" on page 3, and "A new validation study for the importance of the inferred high cytosolic malic enzyme flux in one of the studied cell lines" on page 4.

13. *“The LC-MS methods section presented by the authors are highly problematic. For example they say “Concentrations of metabolites were determined with the standard addition and isotope ratio methods using chemical standards”. This data/information is not presented nor cited. Unclear why they provide upper and lower bounds of metabolite concentrations and not the mean and standard*

deviations of their measurements. Also unclear how absolute concentrations were calculated and if SIL data is actually in a linear and quantifiable range. The methods make no reference to a specific way to calculate this and it is unclear why and which data was collected by several different LC methods. Was a SIM scan the only method used for MS data collection? Or only for specific metabolites? Was a standard curve generated for each SIM scan to calculate absolute concentrations, because absolute concentration ranges are provided? There is no information about AGC values used for analytical measurements and the width of SIM scans is not provided. There is also no specific information about data analysis nor how LC-MS peaks and/or isotopologues were quantified in this study outside of the software that was used. It's difficult for this reviewer to believe that they can accurately detect and quantify lactate isotopologues and the host of other low abundance isotopologues given the methods provided in the manuscript given the methods provided."

We now provide references to the method for quantifying metabolite concentrations using chemical standards and provide all data in a supplementary file.

Two additional methods were used for the determination of compounds that cannot be processed using the SeQuant ZIC-pHILIC column: (i) column SeQuant ZIC-HILIC was used for determination of fumaric acid, alanine, and asparagine; (ii) column Kinetex C18 EVO was used for determination of palmitic acid.

The SIM scan method was used only for the determination of compounds with low concentrations. A detailed list of compounds is provided in the method section.

The width of the ion range for each compound was defined individually in order to collect all isotopic forms within a single range (see Methods).

AGC target was set to 3×10^6 for scan method (72-1080 m/z) and 1×10^6 for SIM measurements (see Methods).

All data analysis for LC-MS peaks and isotopologues were processed using MAVEN software.

For the measurement of low abundant metabolites (or isotopic forms), we concentrated our samples and used a larger injection volume (see Methods).

14. *"The proof that isotope tracing studies were at steady state must be provided in the supplement."*

See "The underlying assumptions of CODE-MFA" - assumption 2 on page 2

15. *"S11 seems to use the wrong mitochondrial pH to calculate Gibbs free energy. There is no reference for why pH 7.5 is used. Most thermodynamic modeling papers have used pH 8.5. This reflects primary biochemical evidence from a number of studies. "*

Mitochondrial pH level estimations range between 7.4–8.2 in different human cell lines^{10–13}. Here, we used mitochondrial pH level of 7.5. We repeated the CODE-MFA analysis for HeLa cells, to verify sensitivity of the results to mitochondrial pH levels of 8 and 8.5. Estimated compartmentalized fluxes were highly similar to those inferred using pH of 7.5, with a pairwise Pearson correlation > 0.98 (Pearson p -value $< 10^{-10}$).

16. *"The authors note in the introduction that a number of papers have used direct measurements of mitochondrial metabolites or have analytically determined the NAD⁺/NADH ratio in cells. However, in the manuscript there is no comparison back to these cited papers about how their algorithm compares. This would be valuable information if the authors want to draw any conclusive, comparative results about their method, and their claims that this algorithm will actually be of utility for the field. The authors do not provide sufficient evidence outside of comparisons to other FBA algorithms to demonstrate method utility or application to published data and exemplary cases. There does not seem to be any effort to relate to information from the field. "*

See "Validation of CODE-MFA compared to state-of-the-art methods and currently known compartmentalized metabolic quantities" on page 3, and "A new validation study for the importance of the inferred high cytosolic malic enzyme flux in one of the studied cell lines" on page 4.

17. *"For figure 1, the reaction diagrams are incorrectly drawn with a one-sided arrow if the authors would like to claim reaction reversibility."*

We changed to two-sided arrows in figure 1a and 1e.

Minor points

18. *Figure legends are chronically mislabeled and inaccurate.*

We thank the reviewer for noticing that. Mislabeled figures were fixed.

19. *"Figure 2 is not insightful. "*

Figure 2-3 were combined with some sub-figures moved to supp. Material.

20. *"The number of significant figures provided in the supplemental tables is incorrect relating values and standard deviations."*

Supplemental table data was fixed.

21. *"The authors note in their introduction that "...we show how mitochondrial and cytosolic metabolic activities can be inferred based on metabolic measurements performed on intact cells under physiological conditions by combining metabolic modelling and computational deconvolution. This claim implies that their method accurately reflects the underlying biology, but they do not test/validate this claim."*

Please see reply regarding method validation on the previous page.

22. "Figure 4 "measured" is actually calculated/inferred"

The asterisks in figure 4d (changed to 3d in the revised manuscript) denote the actual measured total metabolite concentrations.

23. "Why wasn't glucose labeling shown for figure 5? Only glutamine labeling is shown?"

These are just examples of metabolites with a marked difference in isotopic labeling patterns in cytosol and mitochondria. All measured and inferred labeling patterns from both isotopic glucose and glutamine feeding experiments are provided in Tables S2, S3, S4, and S5, for HeLa, HCT116, A549, and LN229 cell lines respectively.

Reviewer #3:

This paper describes a computational approach to deconvolute stable isotope metabolic flux data with the aim to visualise and understand metabolic compartmentalisation between the cytosol and mitochondria. This concept is interesting and novel, and I believe it would be of interest to readers of this journal. I have a few specific comments:

- 1. "My main point is how to validate that the results from the model are correct. Currently the model looks to work well, but there appears to be no hard evidence validating whether it works or not. Could you run the model on multiple different phenotypes and then observe the changes to see how they differ to wild type? This could be picking 3 or so well defined changes that are biochemically well characterised in the literature that have mitochondrial and cytosolic components (e.g. hypoxic response, specific metabolic mutations, chemical metabolic inhibitors). Set some biological data in the lab, then run your model showing the output in a similar format as Figure 5. Then look to see if the biochemical changes you predict are the same as those previously reported in the literature. This would go a long way to showing that the method can accurately predict compartmental metabolic changes. Further to the point above, it would be ideal to show your methodology on less well characterised (and thus novel) and medically relevant changes to see how these alter phenotype. In the discussion you allude to the value of your approach to SHMT1 or IDH mutations. Why not get a biological model with these mutations in the lab and apply your approach to observe and report novel metabolic changes."*

Validating that compartmentalized flux and concentrations inferred by CODE-MFA are correct is tricky, as no other methods enable to do that without cell fractionation that potentially perturbs metabolism – hence, practically, there is no gold-standard to compare with. Studying cells under hypoxia or with mutations in metabolic genes (e.g. IDH1/2) would not enable direct evaluation of compartmentalized flux and concentration quantities, which could not be previously assessed without being biased by cell fractionation.

We evaluate our approach in terms of its ability to reduce the uncertainty in mitochondrial and cytosolic flux and concentration measurements compared to previous methods (FBA and MFA); by comparison to mRNA and protein level measurements of mitochondrial and cytosolic enzymes; and by comparison to redox co-factor ratios measured via bio-sensors (See "Validation of CODE-MFA compared to state-of-the-art methods and currently known compartmentalized metabolic quantities" on page 3).

We chose to focus on applying CODE-MFA to study compartmentalized fluxes in several commonly studied cell lines of different tissue of origin. Following the referee comments, we now employ additional biochemical and genetic approaches to validate one of the model predictions of cell line specific induction and reliance on compartmentalized enzymatic activity – i.e. cytosolic malic enzyme 1 in HCT116 (See "A new validation study for the importance of the inferred high cytosolic malic enzyme flux in one of the studied cell lines" on page 4).

- 2. "The text in paper is complex (particularly the results section), and while I appreciate that it is a complex concept, could you expand some of the results to make it more accessible to the type of scientist who may wish to use this work? i.e. I believe many potential users of this approach will not*

be experts in mathematical modelling, so could these concepts be further explained in a way that is easier to understand? "

We now include short explanation of CODE-MFA for non-experts in mathematical/metabolic modeling in the Results under "A generic approach for inferring mitochondrial and cytosolic metabolite levels and fluxes by computational deconvolution of total cellular concentration and isotope tracing measurements".

3. *"When describing the results in the Hela cells, the pathway figure shown in Figure 5 is useful to the reader. However, this appears to only show one phenotype. What would be really useful is to show multiple phenotypes (see my point above). For example perturb the pathway in a known way (by using a metabolic inhibitor or gene alteration) and then observe the results to see how different they are from the resting phenotype. This will help to show that the model works to describe expected biochemical changes."*

Overlaying compartmentalized flux and concentration data from multiple conditions (cell lines, in our case) in one figure makes it too complex and complicates interpretation. Hence, comparing flux and concentration across cell lines, we show the data in scatter plots in Figure 5 in the revised manuscript.

4. *"It's not totally clear how the first section of the results ("Inferring mitochondrial and cytosolic redox co-factors...") fits with the later sections of the results. Are the NAD/NADH and NADP/NADPH ratios used as an example of how such computational compartmentalisation can be done? Or do these specific reactions form a more fundamental importance for the whole CODE-MFA platform?"*

See response 10 to reviewer #2

5. *"Are you able to show if metabolic reactions in other cellular compartments, e.g. peroxisomes, influence the cytosolic/mitochondrial balance? If these are unaccounted for, could they cause errors in the prediction of cytosolic/mitochondrial balance?"*

Notably, while focusing on cytosolic and mitochondrial metabolism, our analysis is not biased by metabolic activities in nuclei due to free diffusion of small molecules through NPCs into the cytosol³¹; hence, the inferred cytosolic fluxes and metabolite concentrations represents averaged values in cytosol and nucleus. A potential bias in the mitochondrial and cytosolic fluxes and metabolite concentration inference is due to metabolite pools from other organelles such as endoplasmic reticulum, Golgi apparatus, peroxisomes, and lysosomes (as now specified in the Discussion). Focusing on central energy metabolism, we do not expect such major bias from organelles other than mitochondria.

6. *"Considering each metabolic reaction you investigate in the results section, is there is an assumption that this reaction accounts for all the observed change in the metabolites from this reaction? For example in the case of NADPH, the levels appear to be inferred from changes to 6PG & R5P, thus specifically changes to the pentose phosphate pathway. How do you therefore account for changes to*

other processes that either contribute to NADPH/NADP levels, e.g. folate metabolism, fatty acid metabolism etc. If these are missed out from the model could the results you show be incorrect? This comment should also be addressed for the other reactions shown in the paper.”

The flux-force relationship, associating forward-backward flux ratio with reactant concentrations based on thermodynamics, holds for every reaction independently of others. Hence, it is sufficient to model forward-backward flux ration in one redox reaction (as well as reactant concentrations) to infer redox state; and this is how cytosolic NADPH/NADP ratio were evaluated strictly based on the thermodynamics of 6PGD (without having to account for the many other reactions utilizing these redox co-factors).

Reviewer #4:

The idea of directly quantify mitochondrial and cytosolic fluxes and metabolite concentrations via measurements performed with intact cells under physiological conditions (without requiring subcellular fractionation) is new and challenging. However the thermodynamic assumptions and the over simplification of the model in terms of the reactions that are considered as well as on the thermodynamic constraints rise important concerns on the model predictions.

In the following the major concerns identified:

1. *“It is not properly justified that the cytosolic NADP/NADPH balance can be inferred directly from 6PGD activity taking into account that exist other important contributors to the cytosolic NADP/NADPH balance such as NAD⁺ kinase (NADK) (a cytosolic enzyme that can generate NADP and that it has been demonstrated that inhibition of NADK impact NADPH levels Tedeschi PM, Lin H, Gounder M, et al. Suppression of Cytosolic NADPH Pool by Thionicotinamide Increases Oxidative Stress and Synergizes with Chemotherapy. Mol Pharmacol. 2015;88(4):720-727. doi:10.1124/mol.114.096727). ME1 and MTHFD1 are also NADP/NADPH dependent enzymes in the cytosol. “*

The flux-force relationship, associating forward-backward flux ratio with reactant concentrations based on thermodynamics, holds for every reaction independently of others. Hence, it is sufficient to model forward-backward flux ratio in one redox reaction (as well as reactant concentrations) to infer redox state. In the case of NADPH/NADP, we inferred this ratio strictly based on the thermodynamics of 6PGD (without having to account NADK or many other reactions utilizing these redox co-factors): (i) We inferred the relative forward-backward flux through 6PGD by isotope tracing. (ii) We utilized the flux-force relationship to infer Gibbs free energy for 6PGD using the forward-backward flux ratio. (iii) The resulting Gibbs free energy is used to infer cytosolic NADPH/NADP ratio, given the measured cytosolic concentration of other reactants of 6PGD (R5P and 6PG). A similar approach was recently show in Park et al¹⁶. It was also widely used in the past, though without accounting for the displacement from chemical equilibrium^{17,18,20,32}, calculated here using the labeling of 6PG and R5P and the flux-force relationship.

2. *“The authors assume that forward-backward flux through 6PGD can be inferred from M+5 6PG assuming that it is synthesized only through reverse 6PGD flux from R5P (incorporating 12CO₂). Taking into account that 6PGD affinity by CO₂ is very low and that M+5 6PG can be generated from different carbon rearrangements including central metabolism, reversible non-oxidative PP etc... This assumption need to be validated experimentally. “*

M+5 6PG can be synthesized from two sources only: through the oxidative-PPP or through reversed 6PGD from M+5 R5P; hence the employed isotopic balance equation for M+5 6PG is correct and enables to calculate the forward-backward flux through 6PGD. Considering that we measure the relative abundance of the M+5 form of R5P, the above holds regardless of whether the M+5 R5P is synthesized solely through oxidative flux through 6PGD or also via the non-oxidative PPP.

3. *“The central metabolism model not include important reactions that exchange C6, C6 and C3 pools which can have a high impact on mass isotopomers distribution. For instance, the authors not include in the model the non-oxidative branch of PPP which connects pools of C6 with C5, C3..... This pathway need to be included as it is completely reversible and has a high impact on 13C label distribution among central metabolism metabolites.”*

Our work focuses primarily on central energy metabolism through the TCA cycle to demonstrate the unique ability to infer compartmentalized concentration and fluxes. The non-oxidative branch of PPP was left out of this analysis as it is strictly cytosolic, and its activity can be inferred via standard MFA techniques. Not accounting for the non-oxidative branch of PPP had no effect on our results as: (i) We performed tracing with 100% [U-¹³C]-glucose, and hence carbon exchange through the non-oxidative PPP pathway do not affect the labeling of glycolytic intermediates. (ii) Our estimation of forward-backward flux through 6PGD and cytosolic NADPH/NADP ratio is unaffected by not modelling non-oxidative PPP metabolism (see reply to the previous comment).

4. *“In the excel file input.xlsx file (reaction list part of the model inputs), it is written that v2 can only go on the direction Glutamate_media_CY => OTHER_5_carbon being the lower and upper limits 0,1 to 70 . This means that it is assumed that only a flux of entry of glutamate inside the cell is permitted in the model. This is unrealistic taking into account that in many cell models glutamate is mainly released from the cell to the extracellular media. In the .pdf supplementary material file (Table 1) the reaction v2 is written in the reverse direction when compared with the same reaction v2 in the input.xlsx file . The upper and lower limits are both positive which means that according Table 1 v2 can only in the direction cell -> media and in the input.xlsx the same v2 reaction can only go in the reverse direction media->cell which is contradictory.”*

Note that the direction of the glutamate transporter (in input.xlsx) represents flux from cytosol to media. Glutamate_media_CY represents measurements of cytosolic Glutamate using the measurements of Glutamate in the media as a marker for the cytosolic labeling. Therefore, v2 in table 1 and the same v2 in the input.xlsx file have the same direction cell -> media.

5. *“In the excel file input.xlsx (metabolites part of the model input) appear metabolites such as “6phosphogluconate_MT” that not appear in the reaction list. In addition 6-phosphogluconate is only cytosolic and not need to be defined as _MT. There are other examples of metabolites in the input excel (metabolites sheet) that not appear as entities involved in the reaction list. “*

These metabolites exist only in one compartment. For e.g. 6-phosphogluconate is a strictly cytosolic metabolite. In the input.xlsx file I added them in both compartments to simplify the computational process. I constrain them to a very low concentration in the compartment in which they do not exist, practically to represent their inexistence in this compartment.

6. *“The authors infer the cytosolic NAD⁺/NADH ratio by analyzing the forward-backward flux of cytosolic lactate dehydrogenase (LDH), feeding cells with [U-¹³C]-lactate, and measuring the total cellular concentration of pyruvate and lactate (Figure 1e-h). Taking into account that NAD⁺/NADH depends*

on the available substrates and that are other reactions that can contribute to this ratio, this inference need to be further justified or validated. “

The inference of NAD⁺/NADH ratio is strictly based on thermodynamics principles and is not influenced by other reactions that can contribute to this ratio. The forward-backward flux of LDH provides the Gibbs free energy of the reaction, based on the flux-force relationship. The Gibbs free energy is further used, with the concentrations of pyruvate and lactate, to directly compute the cytosolic NAD⁺/NADH ratio (see response above regarding estimating NADPH/NADP ratio). Additionally, the addition of isotopic lactate to media is not expected to change cellular metabolism as the added concentration is low compared to the anyway secreted concentration (See response 4 to reviewer #2).

7. *“Additional evidence is necessary to validate the method. Comparison of the proposed method with other methods in the literature using cell fractionation need to be done. Also it will convenient to compare results with control cells and cells in which key players in the model are knockout to see if the model is able to do an accurate prediction of the decreased flux through the corresponding reaction.”*

See response to 4 to reviewer #1 regarding the comparison with cell fractionation methods.

8. *In the present form the work is not acceptable for publication as additional evidence is necessary to support the conclusions. The assumptions of the model and the reactions considered also need to be justified as important reactions in carbon metabolism that impact in label Distribution are not included in the model.*

See “The underlying assumptions of CODE-MFA” on page 1; and “Validation of CODE-MFA compared to state-of-the-art methods and currently known compartmentalized metabolic quantities” on page 3; and See “A new validation study for the importance of the inferred high cytosolic malic enzyme flux in one of the studied cell lines” on page 4; and Response 1 to Reviewer #1; and Response 3 above.

9. *To reproduce the work it is necessary to have matlab. I do not have matlab in my lab so I cannot try to reproduce. The scripts and files provided seam complete but there is some inconsistencies between Table 1 (supplementary data) and input.xlsx on the directionality of the reactions and the upper and lower bounds used that rise some concerns as detailed in the previous comments. Also it seems inconsistent that in the metabolites sheet (excel imput.xlsx file) appear metabolites that are not used in any reaction.*

See response 4 and 5 above.

References:

1. Boengler, K., Kosiol, M., Mayr, M., Schulz, R. & Rohrbach, S. Mitochondria and ageing: role in heart, skeletal muscle and adipose tissue. *Journal of Cachexia, Sarcopenia and Muscle* Preprint at <https://doi.org/10.1002/jcsm.12178> (2017).
2. Stride, N. *et al.* Decreased mitochondrial oxidative phosphorylation capacity in the human heart with left ventricular systolic dysfunction. *European Journal of Heart Failure* (2013) doi:10.1093/eurjhf/hfs172.
3. Dorn, G. W. Mitochondrial dynamics in heart disease. *Biochimica et Biophysica Acta - Molecular Cell Research* Preprint at <https://doi.org/10.1016/j.bbamcr.2012.03.008> (2013).
4. Milo, R., Jorgensen, P., Moran, U., Weber, G. & Springer, M. BioNumbers The database of key numbers in molecular and cell biology. *Nucleic Acids Research* (2009) doi:10.1093/nar/gkp889.
5. Chen, W. W., Freinkman, E., Wang, T., Birsoy, K. & Sabatini, D. M. Absolute Quantification of Matrix Metabolites Reveals the Dynamics of Mitochondrial Metabolism. *Cell* (2016) doi:10.1016/j.cell.2016.07.040.
6. Posakony, J. W., England, J. M. & Attardi, G. Mitochondrial growth and division during the cell cycle in HeLa cells. *Journal of Cell Biology* (1977) doi:10.1083/jcb.74.2.468.
7. Visser, W. *et al.* Effects of growth conditions on mitochondrial morphology in *Saccharomyces cerevisiae*. *Antonie van Leeuwenhoek* (1995) doi:10.1007/BF00873688.
8. Gerencser, A. A. *et al.* Quantitative measurement of mitochondrial membrane potential in cultured cells: Calcium-induced de- and hyperpolarization of neuronal mitochondria. *Journal of Physiology* (2012) doi:10.1113/jphysiol.2012.228387.
9. Cortese, J. D. Rat liver GTP-binding proteins mediate changes in mitochondrial membrane potential and organelle fusion. *Am J Physiol Cell Physiol* (1999) doi:10.1152/ajpcell.1999.276.3.c611.
10. Haraldsdóttir, H. S., Thiele, I. & Fleming, R. M. T. Quantitative assignment of reaction directionality in a multicompartmental human metabolic reconstruction. *Biophys J* (2012) doi:10.1016/j.bpj.2012.02.032.
11. Porcelli, A. M. *et al.* pH difference across the outer mitochondrial membrane measured with a green fluorescent protein mutant. *Biochem Biophys Res Commun* (2005) doi:10.1016/j.bbrc.2004.11.105.
12. Noor, E., Haraldsdóttir, H. S., Milo, R. & Fleming, R. M. T. Consistent Estimation of Gibbs Energy Using Component Contributions. *PLoS Comput Biol* (2013) doi:10.1371/journal.pcbi.1003098.
13. Sarkar, A. R. *et al.* A ratiometric two-photon probe for quantitative imaging of mitochondrial pH values. *Chem Sci* (2016) doi:10.1039/c5sc03708e.
14. Farré, E. M. *et al.* Analysis of the compartmentation of glycolytic intermediates, nucleotides, sugars, organic acids, amino acids, and sugar alcohols in potato tubers using a nonaqueous fractionation method. *Plant Physiology* (2001) doi:10.1104/pp.010280.

15. Fettke, J., Eckermann, N., Tiessen, A., Geigenberger, P. & Steup, M. Identification, subcellular localization and biochemical characterization of water-soluble heteroglycans (SHG) in leaves of *Arabidopsis thaliana* L.: Distinct SHG reside in the cytosol and in the apoplast. *Plant Journal* (2005) doi:10.1111/j.1365-313X.2005.02475.x.
16. Park, J. O. *et al.* Metabolite concentrations, fluxes and free energies imply efficient enzyme usage. *Nat Chem Biol* **12**, 482–489 (2016).
17. Hedekov, C. J., Capito, K. & Thams, P. Cytosolic ratios of free [NADPH]/[NADP⁺] and [NADH]/[NAD⁺] in mouse pancreatic islets, and nutrient-induced insulin secretion. *Biochem J* **241**, 161–167 (1987).
18. Siess, E. A., Brocks, D. G., Lattke, H. K. & Wieland, O. H. Effect of glucagon on metabolite compartmentation in isolated rat liver cells during gluconeogenesis from lactate. *Biochemical Journal* (1977) doi:10.1042/bj1660225.
19. Veech, R. L., Eggleston, L. V & Krebs, H. a. The redox state of free nicotinamide-adenine dinucleotide phosphate in the cytoplasm of rat liver. *Biochem J* **115**, 609–619 (1969).
20. Krebs, H. A. The redox state of nicotinamide adenine dinucleotide in the cytoplasm and mitochondria of rat liver. *Advances in Enzyme Regulation* (1967) doi:10.1016/0065-2571(67)90029-5.
21. Zhao, Y. *et al.* Genetically encoded fluorescent sensors for intracellular NADH detection. *Cell Metab* (2011) doi:10.1016/j.cmet.2011.09.004.
22. Antoniewicz, M. R., Kelleher, J. K. & Stephanopoulos, G. Determination of confidence intervals of metabolic fluxes estimated from stable isotope measurements. *Metab Eng* **8**, 324–337 (2006).
23. HÄUSSINGER, D. & GEROK, W. Hepatic urea synthesis and pH regulation: Role of CO₂, HCO₃⁻, pH and the activity of carbonic anhydrase. *European Journal of Biochemistry* (1985) doi:10.1111/j.1432-1033.1985.tb09208.x.
24. Lee, W. D., Mukha, D., Aizenshtein, E. & Shlomi, T. Spatial-fluxomics provides a subcellular-compartmentalized view of reductive glutamine metabolism in cancer cells. *Nature Communications* (2019) doi:10.1038/s41467-019-09352-1.
25. Noor, E. *et al.* An integrated open framework for thermodynamics of reactions that combines accuracy and coverage. *Bioinformatics* **28**, 2037–2044 (2012).
26. Itzhak, D. N., Tyanova, S., Cox, J. & Borner, G. H. H. Global, quantitative and dynamic mapping of protein subcellular localization. *eLife* (2016) doi:10.7554/eLife.16950.
27. Nagaraj, N. *et al.* Deep proteome and transcriptome mapping of a human cancer cell line. *Mol Syst Biol* (2011) doi:10.1038/msb.2011.81.
28. Niebel, B., Leupold, S. & Heinemann, M. An upper limit on Gibbs energy dissipation governs cellular metabolism. *Nat Metab* (2019) doi:10.1038/s42255-018-0006-7.
29. Saldida, J. *et al.* Unbiased metabolic flux inference through combined thermodynamic and ¹³C flux analysis. *bioRxiv* Preprint at <https://doi.org/10.1101/2020.06.29.177063> (2020).

30. Bender, T. & Martinou, J. C. The mitochondrial pyruvate carrier in health and disease: To carry or not to carry? *Biochimica et Biophysica Acta - Molecular Cell Research* Preprint at <https://doi.org/10.1016/j.bbamcr.2016.01.017> (2016).
31. Strambio-De-Castillia, C., Niepel, M. & Rout, M. P. The nuclear pore complex: Bridging nuclear transport and gene regulation. *Nature Reviews Molecular Cell Biology* Preprint at <https://doi.org/10.1038/nrm2928> (2010).
32. Veech, R. L., Eggleston, L. V. & Krebs, H. A. The redox state of free nicotinamide-adenine dinucleotide phosphate in the cytoplasm of rat liver. *Biochem J* (1969) doi:10.1042/bj1150609a.

Reviewer #1 (Remarks to the Author):

What are the noteworthy results?

The revised version of the manuscript points that the confidence intervals for compartmentalized concentration and flux estimates from the proposed CODE-MFA are at least 3 magnitudes smaller than those from classical MFA or TMFA.

How does it compare to the established literature? If the work is not original, please provide relevant references.

The estimation of NAD⁺/NADH and NADP⁺/NADPH ratios following similar ideas are already presented in Park et al. but the similarity of the approach is not carefully detailed.

Does the work support the conclusions and claims, or is additional evidence needed?

While the authors have added additional data from different cell lines, the authors do not specifically address the major comments from all reviewers regarding the validation of the estimated compartmentalized concentrations and fluxes; instead, they proceeded to inspect the implications of flux differences between different cell lines, which does not provide firm conclusions. While this required additional amount of work, the question about validating the findings remains unaddressed.

Are there any flaws in the data analysis, interpretation and conclusions? Do these prohibit publication or require revision?

Further inspection of the updated manuscript and supplementary material led to the following major comments:

1. Upon considering the formulation of the objective in CODE-MFA, one sees that simulated quantities do not correspond to measured quantities compared. More specifically, the authors write "The i th convolution of the simulated MIDs of the metabolite in mitochondria and cytosol accounts for the relative pool size of the metabolite in each compartment; calculated by multiplying the relative compartment volume and the simulated metabolite concentration." However, what this corresponds to is the concentration of the MID, i.e. $X_{t,i,j} * C_i$, which is not used in the objective in Eq. (11); instead, only $X_{t,i,j}$ appears as a measured quantity in the second part of this objective. Correction to the objective is required before further inspecting the findings. Note that for correctness, the terms involving C_i^{CY} and C_i^{MT} in second part of the objective should be divided by C_i .

2. Although the response letter points that all chi-square fits are statistically acceptable and directs the reader to the Methods for details, such details do not appear in the updated version of the manuscript. The reviewer is aware that despite allowing for variable concentrations, it is often the case that statistically acceptable fits to the labelling patterns are not easily achievable; thus, the obtained statistics must be presented in the supplement.

3. Ratios of metabolite concentrations can also be estimated from classical TMFA; it would be good to include this analysis in the first section of results, although the reviewer is aware that this would lead to less precise findings. Given the similarity to the approach to that in Park et al. (ref. 39), it would be good to specify the novelty of the approach included in the present work, given that this is the only one which shows consistent results to what is presented in this study.

4. Supplementary Tables 9 and 10 do not include the flux units; that said, it is peculiar that the glutamine uptake flux does not have any error in measurement (lower and upper bounds in Supp. Table 9 are exactly the same). Is there an explanation for this?

Comments 1 and 2, above, require major revision, although that may drastically alter the findings.

Is the methodology sound? Does the work meet the expected standards in your field?

Please, see the comments above.

Is there enough detail provided in the methods for the work to be reproduced?

Code for CODE-MFA is now available and allows reproducibility of the findings.

In addition, the responses to comments from Reviewer #3 were also evaluated as described below.

To comment 1, the reviewer indicate that they make comparison to estimates from FBA, which is not the case. While the authors attempt a validation study, this is done for one enzyme in one cell line, leaving majority of the remaining compartmentalized activities untested.

Comment 2 is fully addressed, as the proposed approach is explained for a general audience in the revised version of the manuscript.

To comment 3, the visualization of Fig. 5 is good since it provides assessing the differences of estimated fluxes between different cell lines. However, this is an issue with the formulation of the approach, that requires careful re-consideration of the implementation and the findings.

Comment 4 is not well addressed, as it does not specify if and to what extent the findings from the model-free conclusions on comparatmentalized pools enter the formulation of the proposed approach dependent on the metabolic model. These details are also not provided in the answer of comment 10 of reviewer #2.

The answer to comment 5 does not provide indication that the limitation of not considering other compartments is the arising issues with the problem formulation (and the largely increasing space of compartmentalized pools that may explain the fitted labeling patterns). The reviewer is not convinced that the findings are not biased by these assumptions.

Finally, point 6 is well addressed.

Reviewer #3 (Remarks to the Author):

I have no further comments

Reviewer #4 (Remarks to the Author):

I carefully read the revised version of this manuscript that the authors submitted and the point-by-point rebuttal letter. I think the authors have done a great effort to handle with the different criticisms raised by the referees. The manuscript has been substantially improved and I think this revised version can be accepted for publication.

We are pleased to submit our revised paper entitled “Inferring mitochondrial and cytosolic metabolism by coupling isotope tracing and deconvolution”, by Stern et al., for publication in Nature Communications. The manuscript has been revised following the reviewer comments, as described below.

Reviewer #1:

Remarks to the author:

What are the noteworthy results?

The revised version of the manuscript points that the confidence intervals for compartmentalized concentration and flux estimates from the proposed CODE-MFA are at least 3 magnitudes smaller than those from classical MFA or TMFA.

CODE-MFA significantly lowers the uncertainty of compartmentalized fluxes, concentrations, and Gibbs free energies compared to existing modelling approaches.

Notably, our approach enables for the first time to infer mitochondrial and cytosolic metabolic activities via measurements performed with intact cells (under physiological conditions), without requiring subcellular fractionation, potentially perturbing metabolic activities.

We show the applicability of the method across a series of proliferating cancer cell lines, enabling to characterize the variability in mitochondrial and cytosolic metabolic activities, undetectable with existing flux inference methods. Specifically, we found major variability in malate-aspartate shuttle flux across cell lines, with a significantly lower flux in HCT116 versus LN229 cell lines; as well as stronger dependence of HCT116 on the cytosolic malic enzyme flux, which was now experimentally validated.

Overall, our study presents a readily usable approach for inferring fluxes and concentrations at a subcellular resolution in Eukaryotic cells. This can be a highly useful tool for studies of metabolic dysfunction in human disease and for metabolic engineering.

How does it compare to the established literature? If the work is not original, please provide relevant references.

The estimation of NAD⁺/NADH and NADP⁺/NADPH ratios following similar ideas are already presented in Park et al. but the similarity of the approach is not carefully detailed.

CODE-MFA provides significantly smaller confidence intervals compared to existing modeling approaches. A variety of methods were proposed for inferring the concentration of metabolites as well as energy and redox cofactor in mitochondria and cytosol, such as performing measurements in isolated mitochondria, and rapid cell fractionation. However, while CODE-MFA strictly relies on measurements performed with intact cells, existing methods potentially perturb the cellular metabolism when extraction specific organelles.

Previous estimates of compartmentalized NAD(P)/NAD(P)H were based on simplified assumptions of enzymes being in chemical equilibrium, and total cellular concentrations of reactants matching the cytosolic/mitochondrial concentrations. As thoroughly described in the paper, CODE-MFA does not rely on such simplifying assumptions in estimating mitochondrial and cytosolic concentrations and redox ratios.

The presented analytical method for estimating the NADP⁺/NADPH ratio indeed follows ideas described by Park et al (and a reference to that work is provided). The analytical estimation of the NAD⁺/NADH ratio was not previously described in this way. Most importantly, our CODE-MFA approach further reduces the uncertainties of redox co-factor concentrations and ratios (compared to the above mentioned, simplified, analytical methods; see Figure 1c-d, 1g-h).

Does the work support the conclusions and claims, or is additional evidence needed?

While the authors have added additional data from different cell lines, the authors do not specifically address the major comments from all reviewers regarding the validation of the estimated compartmentalized concentrations and fluxes; instead, they proceeded to inspect the implications of flux differences between different cell lines, which does not provide firm conclusions. While this required additional amount of work, the question about validating the findings remains unaddressed.

As our method enables for the first time to quantify sub-cellular level metabolite concentration and fluxes without perturbing the cultured cells, there is no gold-standard to compare our inferred quantities. Still, we comprehensively validate our method, as highlighted in our previous response to the Editor – and also provided here:

Validation of CODE-MFA compared to state-of-the-art methods and currently known compartmentalized metabolic quantities

We compared our results for compartmentalized co-factor ratios with previous literature and discuss simplifying assumptions made in literature which biased their estimations. Applied to cultured HeLa cells, the estimated cytosolic NADP⁺/NADPH ratio is consistent with previous measurements performed in iBMK cells using similar thermodynamic considerations⁴. Notably, our estimated NADP⁺/NADPH ratio is

1-2 order of magnitude higher than previous estimates that were based on simplifying assumptions that the NADP⁺/NADPH dependent malic enzyme 1 (ME1)⁵⁻⁷ and isocitrate dehydrogenase 1 (IDH1)⁷ are at chemical equilibrium and that the total cellular concentration of reactants in these enzymes match the cytosolic concentrations; which do not hold here (See response 1 to reviewer #4 in the previous response letter).

We find the cytosolic NAD⁺/NADH ratio is at the upper bound of previous estimates for this ratio⁵; obtained based on thermodynamics of LDH, though without accounting for the displacement from chemical equilibrium; and the mitochondrial NAD⁺/NADH ratio one-order of magnitude lower than previous estimates made based on the thermodynamics beta-hydroxybutyric dehydrogenase⁶⁻⁸, glutamate dehydrogenase^{6,8}, and malate dehydrogenase⁶ (considering similar simplifying assumptions regarding chemical equilibrium of reactions and compartment-specific reactant concentrations). We further compared our inferred compartmentalized NADH levels with those recently reported by Chen et al⁹. We described a straightforward analytical approach for quantifying mitochondrial and cytosolic NADH levels (based on ¹³C-lactate tracing, thermodynamics, and deconvolution), finding that mitochondrial NADH concentration is more than two orders of magnitude higher than in cytosol (Figure 1h). This is consistent with previous measurements performed with an NADH sensor showing ~300-fold higher concentration of NADH in mitochondria than in cytosol¹⁰. However, measurements in purified mitochondria by Chen et al⁹ show 10-fold higher concentration in cytosol than in mitochondria; probably affected by cellular fractionation.

As a complementary approach to validate our inferred compartmentalized fluxes, we now provide additional validation by comparison to mRNA expression and protein level of mitochondrial and cytosolic enzymes (see Figure S4a; Figure S4c).

An additional method for evaluating CODE-MFA is in terms of an ability to uniquely determine mitochondrial and cytosolic flux and concentration compared to previous methods (FBA and MFA). We show that CODE-MFA lowers the uncertainty regarding compartmentalized fluxes and concentrations by one and three orders of magnitude compared to existing modelling approaches, respectively.

In the revised version, we added a validation study of a particularly intriguing finding of substantially high flux through the cytosolic malic enzyme in one of the studied cell lines (HCT116), combining biochemical and genetic approaches (see below).

A new validation study for the importance of the inferred high cytosolic malic enzyme flux in one of the studied cell lines

To demonstrate the applicability of CODE-MFA, we applied it to analyze compartmentalized fluxes on several commonly studied cell lines of different tissue of origin. We found a marked difference in the flux through cytosolic malic enzyme (ME1) between cell lines (Discussion; Table S10).

In the revised manuscript, we now validate the importance of the found high cytosolic malic enzyme flux in HCT116. Considering that glutamine is a major TCA anaplerotic source, we hypothesized that glutamine removal would significantly harm HCT116 having high malic enzyme flux. Indeed, glutamine removal led to >50% drop in cell number after 24 hours; a significantly larger drop than in a control cell line, LN229 (in which we identify lower ME1 flux; Figure L2d, 6d; t-test p-value < 0.03). Consistently, we detect a larger

drop in the concentration of TCA cycle metabolites upon glutamine removal in HCT116 (Figure L2c, 6c; t-test p-value < 0.01). Furthermore, inducible silencing of ME1 leads to a major drop in HCT116 cell proliferation (t-test p-value < 0.01), while silencing of ME1 in LN229 had no effect on cell growth (Figure L2b, 6b). A major contribution of ME1 flux to support NADPH production in HCT116 is supported by a significant drop in NADPH/NADP ratio upon ME1 silencing; while no significant drop in NADPH/NADP ratio is observed upon ME1 silencing in LN229 (Figure L2a, 6a; t-test p-value < 0.01).

Recent studies suggest a variety of mitochondrial and cytosolic specific isozymes promoting cancer specific proliferation, highlighting a growing need for methodologies for probing subcellular level metabolic activities (e.g. IDH1/2, SHMT1/2). Our finding of high flux and importance of ME1 in HCT116 is in line with these studies – though obviously, more work is required to explore ME1 as potential drug target in other colon cell lines and potentially cancers both *in vitro* and *in vivo*. We expect that providing means to directly probe metabolic activities in a subcellular resolution, CODE-MFA, would be highly useful for a variety of studies on metabolic dysfunction in cancer and finding of induced importance of novel drug targets.

Figure L2: Validate the importance of high ME1 flux in HCT116 vs. LN229. (a) NADPH/NADP+ ratio in doxycycline-inducible shRNA ME1 knockdown and a scrambled shRNA (SCR) in HCT116 and LN229 (n=3). (b) Growth (5 days) of doxycycline-inducible shRNA ME1 knockdown and a scrambled shRNA (SCR) in HCT116 and LN229 (n=3). (c) Intracellular relative metabolites intensities in glutamine-free media (n=3). (d) Response of HCT116 and LN229 cells to 24 hours glutamine starvation (n=3).

Are there any flaws in the data analysis, interpretation and conclusions? Do these prohibit publication or require revision?

Further inspection of the updated manuscript and supplementary material led to the following major comments:

1. Upon considering the formulation of the objective in CODE-MFA, one sees that simulated quantities do not correspond to measured quantities compared. More specifically, the authors write “The i th convolution of the simulated MIDs of the metabolite in mitochondria and cytosol accounts for the relative pool size of the metabolite in each compartment; calculated by multiplying the relative compartment volume and the simulated metabolite concentration.” However, what this corresponds to is the concentration of the MID, i.e. $X_{t,i,j} * C_i$, which is not used in the objective in Eq. (11); instead, only $X_{t,i,j}$ appears as a measured quantity in the second part of this objective. Correction to the objective is required before further inspecting the findings. Note that for correctness, the terms involving C_i^{CY} and C_i^{MT} in second part of the objective should be divided by C_i .

We thank the reviewer for pointing our attention to this mistake in the presented equation. We made the mistake when submitting the first revision of the paper (aiming to simplify the method description) – though, importantly, the method was implemented correctly. We have now corrected the mistake in the second part of the objective in equation 11 as following:

$$\sum_{t=1}^K \sum_{i=1}^{N_t} \sum_{j=1}^{N_i} \left(\frac{X_{t,i,j}^{WC} - (\beta_i(C) * X_{t,i,j}^{SIM,CY}(C, v^{net}, \Delta G'^0) + (1 - \beta_i(C)) * X_{t,i,j}^{SIM,MT}(C, v^{net}, \Delta G'^0))}{\sigma_{X_{t,i,j}^{WC}}} \right)^2$$

The convolution of the simulated MIDs of the i th metabolite in mitochondria and cytosol accounts for the relative pool size of the metabolite in each compartment; the relative pool size in the cytosol is denoted β_i and is calculated as following:

$$\beta_i(C) = \frac{\alpha * C_i^{SIM,CY}}{\alpha * C_i^{SIM,CY} + (1 - \alpha) * C_i^{SIM,MT}}$$

where α denotes the relative volume of cytosol in cells.

2. *Although the response letter points that all chi-square fits are statistically acceptable and directs the reader to the Methods for details, such details do not appear in the updated version of the manuscript. The reviewer is aware that despite allowing for variable concentrations, it is often the case that statistically acceptable fits to the labelling patterns are not easily achievable; thus, the obtained statistics must be presented in the supplement.*

We thank the reviewer for noticing that. Goodness of fit description was now also added to the methods.

3. *Ratios of metabolite concentrations can also be estimated from classical TMFA; it would be good to include this analysis in the first section of results, although the reviewer is aware that this would lead to less precise findings. Given the similarity to the approach to that in Park et al. (ref. 39), it would be good to specify the novelty of the approach included in the present work, given that this is the only one which shows consistent results to what is presented in this study.*

Considering the median concentration confidence interval size inferred by TMFA was ~4 orders of magnitude larger than with CODE-MFA (Figure 2h), indeed there is no reason to expect TMFA inferred concentration ratios would be accurate; and hence did not estimate or report that.

See response in page 1 and 2 regarding the novelty of the approach in the present work.

4. *Supplementary Tables 9 and 10 do not include the flux units; that said, it is peculiar that the glutamine uptake flux does not have any error in measurement (lower and upper bounds in Supp. Table 9 are exactly the same). Is there an explanation for this?*

We thank the reviewer for noticing that supplementary Tables 9 and 10 does not include the flux units, and added the units to the table; and also to supplementary Table 1. We added the missing standard deviation for the glutamine uptake measurement to Tables 9 and 10.

In addition, the responses to comments from Reviewer #3 were also evaluated as described below.

To comment 1, the reviewer indicate that they make comparison to estimates from FBA, which is not the case.

We apologize for the mistake. The comparison is made to MFA and not to FBA.

While the authors attempt a validation study, this is done for one enzyme in one cell line, leaving majority of the remaining compartmentalized activities untested.

See above response regarding the extend of validation. We further note the added validation study regarding cell line specific reliance on the cytosolic malic enzyme flux required extensive experimental work, making inducible gene silencing in multiple cell lines and measuring the effect on cellular metabolism and cell proliferation. We think that this additional validation is important to further strengthen the proposed methodology.

Comment 2 is fully addressed, as the proposed approach is explained for a general audience in the revised version of the manuscript.

To comment 3, the visualization of Fig. 5 is good since it provides assessing the differences of estimated fluxes between different cell lines. However, this is an issue with the formulation of the approach, that requires careful re-consideration of the implementation and the findings.

See response #1 above

Comment 4 is not well addressed, as it does not specify if and to what extent the findings from the model-free conclusions on compartmentalized pools enter the formulation of the proposed approach dependent on the metabolic model. These details are also not provided in the answer of comment 10 of reviewer #2.

The model-free analysis merely serves as a demonstration to how the principles of flux/thermodynamics/de-convolution can be easily employed to gain important compartmentalized quantities with a minimal set of assumptions. The inferred quantities are not used in the follow up metabolic model analysis.

The answer to comment 5 does not provide indication that the limitation of not considering other compartments is the arising issues with the problem formulation (and the largely increasing space of compartmentalized pools that may explain the fitted labeling patterns). The reviewer is not convinced that the findings are not biased by these assumptions.

As indicated in our previous reply to comment 5, our analysis may be biased by metabolic activities in compartments others than cytosol and mitochondria; although such a bias is probably minimal if at all, considering that we are focusing on central energy metabolism which is mostly in these compartments.

1. Murai, S. *et al.* Inhibition of malic enzyme 1 disrupts cellular metabolism and leads to vulnerability in cancer cells in glucose-restricted conditions. *Oncogenesis* (2017) doi:10.1038/oncsis.2017.34.
2. Farré, E. M. *et al.* Analysis of the compartmentation of glycolytic intermediates, nucleotides, sugars, organic acids, amino acids, and sugar alcohols in potato tubers using a nonaqueous fractionation method. *Plant Physiology* (2001) doi:10.1104/pp.010280.
3. Fettke, J., Eckermann, N., Tiessen, A., Geigenberger, P. & Steup, M. Identification, subcellular localization and biochemical characterization of water-soluble heteroglycans (SHG) in leaves of *Arabidopsis thaliana* L.: Distinct SHG reside in the cytosol and in the apoplast. *Plant Journal* (2005) doi:10.1111/j.1365-313X.2005.02475.x.
4. Park, J. O. *et al.* Metabolite concentrations, fluxes and free energies imply efficient enzyme usage. *Nat Chem Biol* **12**, 482–489 (2016).
5. Hedeskov, C. J., Capito, K. & Thams, P. Cytosolic ratios of free [NADPH]/[NADP⁺] and [NADH]/[NAD⁺] in mouse pancreatic islets, and nutrient-induced insulin secretion. *Biochem J* **241**, 161–167 (1987).
6. Siess, E. A., Brocks, D. G., Lattke, H. K. & Wieland, O. H. Effect of glucagon on metabolite compartmentation in isolated rat liver cells during gluconeogenesis from lactate. *Biochemical Journal* (1977) doi:10.1042/bj1660225.
7. Veech, R. L., Eggleston, L. V & Krebs, H. a. The redox state of free nicotinamide-adenine dinucleotide phosphate in the cytoplasm of rat liver. *Biochem J* **115**, 609–619 (1969).
8. Krebs, H. A. The redox state of nicotinamide adenine dinucleotide in the cytoplasm and mitochondria of rat liver. *Advances in Enzyme Regulation* (1967) doi:10.1016/0065-2571(67)90029-5.
9. Chen, W. W., Freinkman, E., Wang, T., Birsoy, K. & Sabatini, D. M. Absolute Quantification of Matrix Metabolites Reveals the Dynamics of Mitochondrial Metabolism. *Cell* (2016) doi:10.1016/j.cell.2016.07.040.
10. Zhao, Y. *et al.* Genetically encoded fluorescent sensors for intracellular NADH detection. *Cell Metab* (2011) doi:10.1016/j.cmet.2011.09.004.
11. Strambio-De-Castillia, C., Niepel, M. & Rout, M. P. The nuclear pore complex: Bridging nuclear transport and gene regulation. *Nature Reviews Molecular Cell Biology* Preprint at <https://doi.org/10.1038/nrm2928> (2010).

Reviewer #1 (Remarks to the Author):

The reviewer would like to thank the authors for carefully considering the comments from the last revision -- particularly the comment regarding the optimized objective.

There is one major issue remaining -- all code has to be provided along with the publication, since currently there is no way for the reviewer to check that the implementation is as specified in the problem formulation.

With this issue resolved, the reviewer agrees that the proposed approach provides an improvement over the standard MFA approach; however, it still remains to be shown that predictions about compartmentalized pool sizes match measurements (from techniques to be developed).

Minor comment

Fig. 6 could be improved (e.g. remove the gray lines, improve the naming of the lines / modifications / treatments). Along these lines, the sentence starting with "Validate the importance of high ME1 flux in HCT116 ... " needs improvement; probably you want to say "To validate ... ".

There is one major issue remaining -- all code has to be provided along with the publication, since currently there is no way for the reviewer to check that the implementation is as specified in the problem formulation.

CODE-MFA code is available at the following link:

<https://github.com/sternal75/CODE-MFA>

Link to the code is also provided in the code availability section within the manuscript.

With this issue resolved, the reviewer agrees that the proposed approach provides an improvement over the standard MFA approach; however, it still remains to be shown that predictions about compartmentalized pool sizes match measurements (from techniques to be developed).

Minor comment:

Fig. 6 could be improved (e.g. remove the gray lines, improve the naming of the lines / modifications / treatments).

We thank the reviewer for this comment, and improved Fig. 6 according the above.

Along these lines, the sentence starting with "Validate the importance of high ME1 flux in HCT116 ... " needs improvement; probably you want to say "To validate ... ".

Text was changed accordingly.